# IMPROVING OUT-OF-DISTRIBUTION ROBUSTNESS VIA SELECTIVE AUGMENTATION

## ABSTRACT

Machine learning algorithms typically assume that training and test examples are drawn from the same distribution. However, distribution shifts is a common problem in real-world applications and can cause models to perform dramatically worse at test time. In this paper, we specifically consider the problems of domain shifts and subpopulation shifts, where learning invariant representations by aligning domain-specific representations or balancing the risks across domains with regularizers are popular solutions. However, designing regularizers that are suitable for diverse real-world datasets is challenging. Instead, we shed new light on addressing distribution shifts by directly eliminating domain-related spurious correlations with augmentation, leading to a simple technique based on mixup, called LISA (Learning Invariant Representations via Selective Augmentation). LISA selectively interpolates samples either with the same labels but different domains or with the same domain but different labels. Empirically, we study the effectiveness of LISA on nine benchmarks ranging from subpopulation shifts to domain shifts. The results indicate that LISA consistently outperforms other state-of-the-art methods with superior invariant representations. The empirical findings are further strengthened by our theoretical analysis.

## 1 INTRODUCTION

To deploy machine learning algorithms in real-world applications, we must pay attention to distribution shifts, i.e. when the test distribution is different from the training distribution, which substantially degrades model performance. In this paper, we refer this problem as out-of-distribution (OOD) generalization and specifically consider performance gaps caused by two kinds of distribution shifts: *domain shifts* and *subpopulation shifts*. In domain shifts, the test data is sampled from different domains than the training data, which requires the trained model to generalize well to test domains without seeing training data from those domains. Take health risk prediction as an example. We may want to train a model on patients from a few sampled hospitals and then deploy the model to a broader set of hospitals (Koh et al., 2021). In subpopulation shifts, the proportions of subpopulations in the test distribution differ from the proportions in the training distribution. When subpopulation shift occur, models perform poorly when they falsely rely on spurious correlations, which may occur when some subpopulations are under-represented in the training set. For example, in financial risk prediction, a machine learning model trained on the entire population may associate the labels with demographic features (e.g., religion and race), making the model fail on the test set when such an association does not hold in reality.

To improve model robustness under these two kinds of distribution shifts, methods for learning invariant representations have shown effectiveness in various applications. These methods learn features or prediction mechanisms that are invariant to different domains while still containing sufficient information for the targeted task (Li et al., 2018; Arjovsky et al., 2019). Concretely, some prior works learn invariant representations by aligning and regularizing the domain-specific representations (Li et al., 2018; Sun & Saenko, 2016). Other works aim to find invariant representations by balancing the risk across domains using regularizers (Arjovsky et al., 2019; Krueger et al., 2021; Rosenfeld et al., 2021), which further increases the dependency between the invariant representations and labels. However, designing regularizers that are widely suitable to datasets from diverse domains is especially challenging and insuitable regularizers may adversely limit the model's expressive power, leading to inconsistent performance among various real-world datasets. For example, on the WILDS datasets (Koh et al., 2021), invariant risk minimization (IRM) (Arjovsky et al.,

2019) outperforms empirical risk minimization (ERM) on CivilComments, but fails to improve robustness on a variety of other datasets like Camelyon17 and RxRx1. A similar phenomenon is also reflected in the performance of CORAL (Sun & Saenko, 2016).

Instead of explicitly imposing regularization to learn invariant representations, we turn towards an implicit solution. Inspired by mixup (Zhang et al., 2018), we aim to alleviate the effects of domain-related spurious information through data interpolation, leading to a simple algorithm called **LISA** (**L**earning **I**nvariant Representations with **S**elective **A**ugmentation). Concretely, LISA linearly interpolates the features for a pair of samples and applies the same interpolation strategy on the corresponding labels. Critically, the pairs are selectively chosen according to two sample selection strategies. In selection strategy I, LISA interpolates samples with the same label but from different domains, aiming to eliminate domain-related spurious correlations. In selection strategy II, LISA interpolates samples with the same domain but different labels, where the model should to ignore the domain information and generate different predicted values as the interpolation ratio changes. In this way, LISA encourages the model to learn domain-invariant predictors without explicitly constraining or regularizing the representation.

The primary contributions of this paper are as follows: (1) We develop a method that tackles the problem of distribution shifts by canceling out the domain-related spurious correlations via data interpolation. (2) We conduct broad empirical experiments to evaluate the effectiveness of LISA on nine benchmark datasets from diverse domains. In these experiments, we make the following observations. First, we find that LISA consistently outperforms seven prior methods in addressing both domain shifts and subpopulation shifts. Second, we identify that the performance gains of LISA are indeed caused by canceling out domain-specific information and learning invariant representations, rather than simply involving more data via interpolation. Third, when the degree of distribution shift increases, LISA achieves more significant performance gains. (3) Finally, we provide theoretical analysis of the phenomena distilled from the empirical studies, where we provably demonstrate that LISA can achieve smaller worst-domain error compared with ERM and vanilla mixup. We also note that to the best of our knowledge, this is the first theoretical analysis of how mixup (with or without the selection strategies) affects mis-classification error.

## 2 PRELIMINARIES

In this paper, we consider the setting where one predicts the label $y \in \mathcal{Y}$ based on the input feature $x \in \mathcal{X}$. Given a parameter space $\Theta$ and a loss function $\ell$, we are supposed to train a model $f_\theta$ under the training distribution $P_{tr}$, where $\theta \in \Theta$. In empirical risk minimization (ERM), assume the empirical distribution over training data is $\hat{P}_{tr}$, ERM optimizes the following objective:

$$\theta^* := \arg\min_{\theta \in \Theta} \mathbb{E}_{(x,y) \sim \hat{P}}[\ell(f_\theta(x), y)]. \quad (1)$$

In a traditional machine learning setting, a test set, sampled from a test distribution $P_{ts}$, is used to evaluate the generalization of the trained model $\theta^*$, where the test distribution is assumed to be the same as the training distribution, i.e., $P^{tr} = P^{ts}$. In this paper, we are interested in the setting when distribution shift occurs, i.e., $P^{tr} \neq P^{ts}$.

Specifically, follow Muandet et al. (2013); Albuquerque et al. (2019); Koh et al. (2021), we regard the overall data distribution containing $\mathcal{D} = \{1, \dots, D\}$ domains and each domain $d \in \mathcal{D}$ is associated with a data distribution $P_d$ over a set $(X, Y, d) = \{(x_i, y_i, d)\}_{i=1}^{N^d}$, where $N^d$ is the number of samples in domain $d$. Then, we formulate the training distribution as the mixture of $D$ domains,

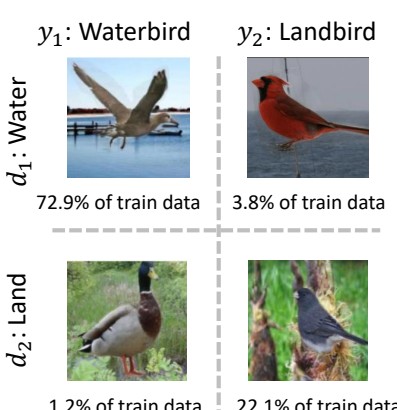

$y_1$: Waterbird  $y_2$: Landbird

$d_1$: Water

72.9% of train data | 3.8% of train data

$d_2$: Land

1.2% of train data | 22.1% of train data

Figure 1: Group illustration of Waterbird data. Domains and labels are represented by background and bird type. Most training samples are drawn from waterbird in water ($d_1$, $y_1$) and landbird in land ($d_2$, $y_2$). The trained model is spuriously biased by the background.

i.e., $P^{tr} = \sum_{d \in \mathcal{D}} r_d^{tr} P_d$, where $\{r_d^{tr}\}$ denotes the mixture probabilities in training set. Here, the training domains are defined as $\mathcal{D}^{tr} = \{d \in \mathcal{D} | r_d^{tr} > 0\}$. Similarly, the test distribution could be represented as $P^{ts} = \sum_{d \in \mathcal{D}} r_d^{ts} P_d$, where $\{r_d^{ts}\}$ is the mixture probabilities in test set. The test domains are defined as $\mathcal{D}^{ts} = \{d \in \mathcal{D} | r_d^{ts} > 0\}$.

In domain shifts, we investigate the problem that the test domains are disjoint from the training domains, i.e., $\mathcal{D}^{tr} \cap \mathcal{D}^{ts} = \emptyset$. In general, we assume the test domains share some common properties with the training domains. For example, in Camelyon17 (Koh et al., 2021) data, we train the model on some hospitals and test it in a new hospital. We evaluate the worst-domain or average performance of the classifier among all test domains.

In subpopulation shifts, the test set have domains that have been seen in the training set, but with a different proportion of subpopulations, i.e., $\mathcal{D}^{ts} \subseteq \mathcal{D}^{tr}$ but $\{r_d^{ts}\} \neq \{r_d^{tr}\}$. Under this setting, follow Sagawa et al. (2020a), we specially consider group-based spurious correlations, where each group $g \in \mathcal{G}$ is defined to be associated with a domain $d$ and a label $y$, i.e., $g = (d, y)$. We assume the domain identification is spuriously correlated with the label. For example, we illustrate the Waterbirds dataset in Figure 1, where the background $d$ (water or land) is spuriously correlated with the label $y$ (waterbird or landbird). Based on the group definition, we evaluate the model via the worst test group error, i.e., $\max_g \mathbb{E}_{(x,y) \sim g}[\ell_{0-1}(f_\theta(x), y)]$, where $\ell_{0-1}$ represents the 0-1 loss.

# 3 LEARNING INVARIANT REPRESENTATIONS WITH SELECTIVE AUGMENTATION

This section presents LISA, a simple way to improve robustness to domain shifts or subpopulation shifts. The key idea behind LISA is to encourage the model to alleviate the effects of domain-related spurious correlations by selective data interpolation. Before detailing how to select interpolated samples, we first provide a general formulation for data interpolation.

In LISA, we perform linear interpolation between training samples. Specifically, given samples $(x_i, y_i, d_i)$ and $(x_j, y_j, d_j)$ drawn from domains $d_i$ and $d_j$, we apply mixup (Zhang et al., 2018), a simple data interpolation strategy, separately on the input features and corresponding labels as:

$$x_{mix} = \lambda x_i + (1 - \lambda)x_j, \ y_{mix} = \lambda y_i + (1 - \lambda)y_j, \quad (2)$$

where the interpolation ratio $\lambda \in [0, 1]$ is sampled from a Beta distribution $\text{Beta}(\alpha, \beta)$ and $y_i$ and $y_j$ are one-hot vectors for classification problem. Notice that the mixup approach in equation 2 can be replaced by CutMix (Yun et al., 2019), which shows stronger empirical performance in vision-based applications. In text-based applications, we replace the feature interpolation in equation 2 with Manifold Mixup (Verma et al., 2019), where the interpolation strategy is performed on the output representation of a pre-trained model, e.g., the output of BERT (Devlin et al., 2019).

After obtaining the interpolated features and labels, we replace the original features and labels in ERM with the interpolated ones. Then, the optimization process in equation 1 is reformulated as:

$$\theta^* := \arg\min_{\theta \in \Theta} \mathbb{E}_{\{(x_i, y_i, d_i), (x_j, y_j, d_j)\} \sim \hat{P}}[\ell(f_\theta(x_{mix}), y_{mix})]. \quad (3)$$

Without additional selection strategies, vanilla mixup will regularize the model and reduce overfitting (Zhang et al., 2021b), allowing it to attain good in-distribution generalization. However, vanilla mixup may not be able to cancel out spurious correlations, causing the model to still fail at attaining good OOD generalization (see empirical comparisons in Section 4.3 and theoretical discussion in Section 5). In LISA, we instead adopt a new strategy where mixup is only applied across specific domains or groups, which leans towards learning invariant representations and thus better OOD performance. Specifically, the two kinds of sample selection strategies are presented as follows:

- **Selection Strategy I: Interpolating samples with the same label.** In selection strategy I, LISA interpolates samples with the same label but different domains (i.e., $d_i \neq d_j$, $y_i = y_j$). This produces datapoints that have both domains partially present, effectively eliminating spurious correlations between domain and label in cases where the pair of domains correlate differently with the label. Additionally, if domain information does not fully reflect the spurious correlations in some datasets, we can also enlarge the interpolation scope to cover more potentially spurious correlations by only interpolating samples within the same class regardless domain information (i.e., $y_i = y_j$). As a result, LISA with selection strategy I should learn domain-invariant representations for each class and thus achieve better OOD robustness.

- **Selection Strategy II: Interpolating samples with the same domain.** Supposing domain information is highly spuriously correlated with the label information, selection strategy II applies the

interpolation strategy on samples with the same domain but different labels, i.e., $d_i = d_j, y_i \neq y_j$. Intuitively, even within the same domain, the model is supposed to generate different predicted labels since the interpolation ratio $\lambda$ is randomly sampled, corresponding to different labels $y_{mix}$. This causes the model to make predictions that are less dependent on the domain, again improving OOD robustness.

In LISA, we randomly perform selection strategies I or II during the training process with probability $p_{sel}$ and $1 - p_{sel}$ for each batch of data, where $p_{sel}$ is treated as a hyperparameter in our experiments. The choice of $p_{sel}$ depends on the number of domains and the relation between domain information and spurious correlations. Empirically, using selection strategy I only brings much more benefits when there are more domains, and/or the domain information can not fully reflect the spurious correlations. LISA with selection strategy II can benefit the performance when domain information is highly spuriously correlated with the label, where we find a balanced ratio (i.e., $p_{sel} = 0.5$) performs the best. The pseudocode of the training procedure of LISA is shown in Algorithm 1.

---

**Algorithm 1** Training Procedure of LISA

---

**Require:** Training data, Step size $\eta$
1: **while** not converge **do**
2:     Sample $\lambda \sim \text{Beta}(\alpha, \beta)$
3:     Sample a set of samples $B_1$ uniformly from the training data
4:     Randomly select a sample selection strategy I or II with the probability $p_{sel}$ and $1 - p_{sel}$
5:     **if** use selection strategy I **then**
6:         For each sample $(x_i, y_i, d_i)$, find another one $(x_j, y_j, d_j)$ from the dataset with the same label $(y_i = y_j)$ but different domains $(d_i \neq d_j)$, and construct set $B_2$.
7:     **else if** use selection strategy II **then**
8:         For each sample $(x_i, y_i, d_i)$, find another one $(x_j, y_j, d_j)$ from the same domain $(d_i = d_j)$ but different labels $(y_i \neq y_j)$, constructing set $B_2$.
9:     Update $\theta$ with data $\lambda B_1 + (1 - \lambda)B_2$.

---

## 4 EXPERIMENTS

In this section, we conduct comprehensive experiments to evaluate the effectiveness of LISA. Specifically, we aim to answer the following questions: **Q1**: Compared to prior methods, can LISA improve robustness to domain shifts and subpopulation shifts (Section 4.1 and Section 4.2)? **Q2**: Which aspects of LISA are the most important for improving robustness (Section 4.3)? **Q3**: How does LISA perform with varying degrees of distribution shifts (Section 4.4)? **Q4**: Does LISA successfully produce invariant representations (Section 4.5)?

To answer Q1, we compare to ERM, IRM (Arjovsky et al., 2019), MMD (Li et al., 2018), DRNN (Ganin & Lempitsky, 2015), GroupDRO (Sagawa et al., 2020a), DomainMix (Xu et al., 2020), and Fish (Shi et al., 2021). Upweighting (UW) is particularly suitable for subpopulation shifts, so we also use it for comparison. We adopt the same model architectures for all approaches.

### 4.1 EVALUATING ROBUSTNESS TO DOMAIN SHIFTS

**Experimental Setup.** To study domain shifts, we study five datasets. Four datasets (Camelyon17, FMoW, RxRx1, and Amazon) are selected from WILDS (Koh et al., 2021), covering real-world distribution shifts across diverse domains (e.g., health, natural language process, and vision). Besides the WILDS data, we also apply LISA on the MetaShift datasets (Liang & Zou, 2021), constructed using the real-world images and natural heterogeneity of Visual Genome (Krishna et al., 2016). We summarize the datasets in Table 8 of Appendix A.1.1, including domain information, evaluation metric, model architecture, and the number of classes. Detailed dataset descriptions and other training details are discussed in Appendix A.1.1 and A.1.2, respectively.

**Results.** We report the results of domain shifts in Table 1, where the full results that include validation performance and other metrics are listed in Appendix A.1.3. The optimal strategy selection probability $p_{sel}$ is set as 1.0 for these domain shifts problems, i.e., only selection strategy I is used.

In addition, we only interpolate samples with the same labels regardless the domain information, which empirically leads to the best performance. According to Table 1, we have two key findings:

- There are no significant performance differences between ERM and other invariant representation learning methods (e.g. IRM, CORAL, DomainMix) on most datasets, which is consistent with the reported results on WILDS (Koh et al., 2021). A potential reason is that the existing domain information may not fully reflects the spurious correlation. For example, in Camelyon17-wilds dataset (Koh et al., 2021), the presence of tumor tissue (i.e., label) mainly depends on the demographic of patients (e.g., race, gender), which shows no significant difference across hospitals (i.e., domain information). This could also explain why only interpolating samples with the same labels regardless the domain information achieves the best performance, which also corroborates our claims in the description of the selection strategy I in Section 3.

- The consistent superiority of LISA outperforms prior methods on all five datasets regardless of the model architecture and dataset types (i.e., image or text), demonstrating its effectiveness in improving OOD robustness by canceling out spurious correlations with augmentation.

Table 1: Main domain shifts results. LISA outperforms prior methods on all five datasets. Following the instructions of Koh et al. (2021), we report the performance of Camelyon17 over 10 different seeds and the results of other datasets are obtained over 3 different seeds.

|  | Camelyon17 | FMoW | RxRx1 | Amazon | MetaShift |
|---|---|---|---|---|---|
|  | Avg. Acc. | Worst Acc. | Avg. Acc. | 10-th Per. Acc. | Worst Acc. |
| ERM | $70.3 \pm 6.4\%$ | $32.3 \pm 1.25\%$ | $29.9 \pm 0.4\%$ | $53.8 \pm 0.8\%$ | $52.1 \pm 0.4\%$ |
| IRM | $64.2 \pm 8.1\%$ | $30.0 \pm 1.37\%$ | $8.2 \pm 1.1\%$ | $52.4 \pm 0.8\%$ | $51.8 \pm 0.8\%$ |
| CORAL | $59.5 \pm 7.7\%$ | $31.7 \pm 1.24\%$ | $28.4 \pm 0.3\%$ | $52.9 \pm 0.8\%$ | $47.6 \pm 1.9\%$ |
| GroupDRO | $68.4 \pm 7.3\%$ | $30.8 \pm 0.81\%$ | $23.0 \pm 0.3\%$ | $53.3 \pm 0.0\%$ | $51.9 \pm 0.7\%$ |
| DomainMix | $69.7 \pm 5.5\%$ | $34.2 \pm 0.76\%$ | $30.8 \pm 0.4\%$ | $53.3 \pm 0.0\%$ | $51.3 \pm 0.5\%$ |
| Fish | $74.7 \pm 7.1\%$ | $34.6 \pm 0.18\%$ | $10.1 \pm 1.5\%$ | $53.3 \pm 0.0\%$ | $49.2 \pm 2.1\%$ |
| **LISA (ours)** | $\mathbf{77.1 \pm 6.5\%}$ | $\mathbf{35.5 \pm 0.65\%}$ | $\mathbf{31.9 \pm 0.8\%}$ | $\mathbf{54.7 \pm 0.0\%}$ | $\mathbf{54.2 \pm 0.7\%}$ |

## 4.2 EVALUATING ROBUSTNESS TO SUBPOPULATION SHIFTS

**Evaluation Protocol.** In subpopulation shifts, we evaluate the performance on four binary classification datasets, including Colored MNIST (CMNIST), Waterbirds (Sagawa et al., 2020a), CelebA (Liu et al., 2015), and Civilcomments (Borkan et al., 2019). We summarize brief data statistics in Table 14 of Appendix A.2.1, covering domain information, model architecture, and class information. Full dataset descriptions of subpopulation shifts are also presented in Appendix A.2.1. Following Sagawa et al. (2020a), in subpopulation shifts, we use the worst-group accuracy to evaluate the performance of all approaches and the domain identifications are highly spurious correlated with the label information. For example, as suggested in Figure 1, 95% images in the Waterbirds dataset have the same background and bird type, i.e., waterbirds in water or landbirds in land. Full hyperparameter settings and training details are listed in Appendix A.2.2.

**Results.** In Table 2, we report the overall performance of LISA and other methods. Similar to the observations in domain shifts, LISA consistently outperforms prior methods in CMNIST, CelebA, and CivilComments. In Waterbirds, LISA outperforms other invariant representation learning methods (e.g., IRM, CORAL, DomainMix, Fish) and shows similar performance to GroupDRO. These results demonstrate the effectiveness of LISA in improving OOD robustness. In CMNIST, Waterbirds, and CelebA, we find that $p_{sel} = 0.5$ works well for choosing selection strategies I and II, while $p_{sel}$ is set as $1.0$ in CivilComments. This is not surprising because it might be more beneficial to use the strategy I more often to eliminate domain effects when there are more domains, i.e., eight domains in CivilComments v.s. two domains in others.

## 4.3 ABLATION STUDY: IS THE PERFORMANCE GAIN FROM DATA AUGMENTATION?

In LISA, we apply selective interpolation strategies on samples either with the same label but different domains or with the same domain but different labels. Here, we explore two substitute interpolation strategies: (1) *Vanilla mixup*: In Vanilla mixup, we do not add any constraint on the

Table 2: Results of subpopulation shifts. Here, we show the average and worst group accuracy. We repeat the experiments three times and put full results with standard deviation in Table 16.

| | CMNIST Avg. | CMNIST Worst | Waterbirds Avg. | Waterbirds Worst | CelebA Avg. | CelebA Worst | CivilComments Avg. | CivilComments Worst |
|---|---|---|---|---|---|---|---|---|
| ERM | 27.8% | 0.0% | 97.0% | 63.7% | 94.9% | 47.8% | 92.2% | 56.0% |
| UW | 72.2% | 66.0% | 95.1% | 88.0% | 92.9% | 83.3% | 89.8% | 69.2% |
| IRM | 72.1% | 70.3% | 87.5% | 75.6% | 94.0% | 77.8% | 88.8% | 66.3% |
| CORAL | 71.8% | 69.5% | 90.3% | 79.8% | 93.8% | 76.9% | 88.7% | 65.6% |
| GroupDRO | 72.3% | 68.6% | 91.8% | **90.6%** | 92.1% | 87.2% | 89.9% | 70.0% |
| DomainMix | 51.4% | 48.0% | 76.4% | 53.0% | 93.4% | 65.6% | 90.9% | 63.6% |
| Fish | 46.9% | 35.6% | 85.6% | 64.0% | 93.1% | 61.2% | 89.8% | 71.1% |
| **LISA (ours)** | 74.0% | **73.3%** | 91.8% | 89.2% | 92.4% | **89.3%** | 89.2% | **72.6%** |

sample selection, i.e., the mixup is performed on any pairs of samples; (2) *In-group mixup*: This strategy applies data interpolation on samples with the same labels and from the same domains. Notice that all the substitute interpolation strategies use the same mixup types (e.g., mixup/Manifold Mixup/CutMix) as LISA. Finally, since upweighting (UW) small groups significantly improves performance in subpopulation shifts, we also evaluate UW combined with Vanilla/In-group mixup.

The ablation results of domain shifts and subpopulation shifts are in Table 3 and Table 4, respectively. Furthermore, we also conduct experiments on datasets without spurious correlation in Table 18 of Appendix A.3. From the results, we make the following three key observations. First, compared with Vanilla mixup, the performance of LISA verifies that selective data interpolation does improve the out-of-distribution robustness by canceling out the spurious correlations and encouraging invariant representation learning rather than simply data augmentation. This findings are further strengthened by the results in Table 18, where Vanilla mixup outperforms LISA and ERM without spurious correlations but LISA achieves the best performance with spurious correlations. Second, the superiority of LISA over In-group mixup verifies that only interpolating samples within each group is incapable of eliminating out the spurious information, where In-group mixup still performs the role of data augmentation. Finally, though incorporating UW significantly improves the performance of Vanilla mixup and In-group mixup in subpopulation shifts, LISA still achieves larger benefits than these enhanced substitute strategies, demonstrating its stronger power in improving OOD robustness.

Table 3: Compared LISA with substitute mixup strategies in domain shifts.

| | Camelyon17 Avg. Acc. | FMoW Worst Acc. | RxRx1 Avg. Acc. | Amazon 10-th Per. Acc. | MetaShift Worst Acc. |
|---|---|---|---|---|---|
| ERM | $70.3 \pm 6.4\%$ | $32.8 \pm 0.45\%$ | $29.9 \pm 0.4\%$ | $53.8 \pm 0.8\%$ | $52.1 \pm 0.4\%$ |
| Vanilla mixup | $71.2 \pm 5.3\%$ | $34.2 \pm 0.45\%$ | $26.5 \pm 0.5\%$ | $53.3 \pm 0.0\%$ | $51.3 \pm 0.7\%$ |
| In-group mixup | $75.5 \pm 6.7\%$ | $32.2 \pm 1.18\%$ | $24.4 \pm 0.2\%$ | $53.8 \pm 0.6\%$ | $52.7 \pm 0.5\%$ |
| **LISA (ours)** | $\mathbf{77.1 \pm 6.5\%}$ | $\mathbf{35.5 \pm 0.65\%}$ | $\mathbf{31.9 \pm 0.8\%}$ | $\mathbf{54.7 \pm 0.0\%}$ | $\mathbf{54.2 \pm 0.7\%}$ |

Table 4: Compared LISA with substitute mixup strategies in subpopulation shifts. UW represents upweighting. Full results with standard deviation is listed in Table 17.

| | CMNIST Avg. | CMNIST Worst | Waterbirds Avg. | Waterbirds Worst | CelebA Avg. | CelebA Worst | CivilComments Avg. | CivilComments Worst |
|---|---|---|---|---|---|---|---|---|
| ERM | 27.8% | 0.0% | 97.0% | 63.7% | 94.9% | 47.8% | 92.2% | 56.0% |
| Vanilla mixup | 32.6% | 3.1% | 81.0% | 56.2% | 95.8% | 46.4% | 90.8% | 67.2% |
| Vanilla mixup + UW | 72.2% | 71.8% | 92.1% | 85.6% | 91.5% | 88.0% | 87.8% | 66.1% |
| In-group mixup | 33.6% | 24.0% | 88.7% | 68.0% | 95.2% | 58.3% | 90.8% | 69.2% |
| In-group mixup + UW | 72.6% | 71.6% | 91.4% | 87.1% | 92.4% | 87.8% | 84.8% | 69.3% |
| **LISA (ours)** | 74.0% | **73.3%** | 91.8% | **89.2%** | 92.4% | **89.3%** | 89.2% | **72.6%** |

## 4.4 EFFECT OF THE DEGREE OF DISTRIBUTION SHIFTS

We further investigate the performance of LISA with respect to the degree of distribution shifts. Here, we use MetaShift to evaluate performance, where the distance between training and test domains is measured as the node similarity on a metagraph (Liang & Zou, 2021). To vary the distance between training and test domains, we change the backgrounds of training objects (see full experimental details in Appendix A.1.1). The performance with varied distances is illustrated in Table 5, where the top four best methods (i.e., ERM, Group-DRO, IRM, DomainMix) are reported for compar-

Table 5: Effects of the degree of distribution shifts w.r.t. the performance. Distance represents the distribution distance between training and test domains.

| Distance | 0.44 | 0.71 | 1.12 | 1.43 |
|---|---|---|---|---|
| ERM | 80.1% | 68.4% | 52.1% | 33.2% |
| IRM | 79.5% | 67.4% | 51.8% | 32.0% |
| DomainMix | 76.0% | 63.7% | 51.3% | 30.8% |
| GroupDRO | 77.0% | 68.9% | 51.9% | 34.2% |
| **LISA (ours)** | **81.3%** | **69.7%** | **54.2%** | **37.5%** |

ison. We observe that LISA consistently outperforms other methods under all scenarios. Additionally, another interesting finding is that LISA achieves more substantial improvements with the increases of distance. A potential reason is that the models may rely more heavily on domain correlations when there is a larger distance between training and test domains.

## 4.5 Analysis about Learned Invariance

Finally, we analyze the invariance learned by LISA. Specifically, we investigate representation-level and prediction-level invariances below.

**Representation-level Invariance.** For each label $y$, assume the hidden representation for each domain $d$ as $H_d^y$. The representation-level invariance $\mathrm{IV}_r$ is measured by the pairwise KL divergence of distribution $P(H_d^y)$ among all domains as $\mathrm{IV}_r = \frac{1}{|\mathcal{Y}||\mathcal{D}|^2} \sum_{y \in \mathcal{Y}} \sum_{d',d \in \mathcal{D}} \mathrm{KL}(P(H_D^y \mid D = d)|P(H_D^y \mid D = d'))$, where smaller $\mathrm{IV}_r$ values indicate that the learned representations are more invariant with respect to the labels. We report the results on CMNIST and MetaShift in Table 6. Our key observations are: (1) Com-

Table 6: Results of representation-level invariance ($\times 10^8$), where smaller values denote stronger invariance w.r.t. labels.

| | CMNIST | MetaShift |
|---|---|---|
| ERM | 1.683 | 0.632 |
| Vanilla mixup | 4.392 | 0.634 |
| IRM | 1.905 | 0.627 |
| DomainMix | 2.155 | 0.614 |
| **LISA (ours)** | **0.421** | **0.585** |

pared with ERM, the invariance of LISA indicates its promise in improving the OOD robustness by encouraging invariant representation learning. (2) LISA has greater invariance than vanilla mixup, validating that the invariant representations are not caused by naive data interpolation. (3) LISA provides more invariant representations than regularization-based methods, i.e., IRM and DomainMix.

**Prediction-level Invariance.** Motivated by Arjovsky et al. (2019); Krueger et al. (2021), the prediction-level invariance $\mathrm{IV}_p$ is measured by the variance of test risks across all domains, i.e., $\mathrm{IV}_p = \mathrm{Var}(\{\mathcal{R}_1(\theta), \ldots, \mathcal{R}_D(\theta)\})$, where $D$ represents the number of test domains. A small $\mathrm{IV}_p$ represents strong prediction-level invariance. The results on CMNIST and MetaShift are reported in Table 7, where the superiority of LISA verifies that it could also improve the prediction-level invariance.

Table 7: Results of the analysis of prediction-level invariance. Smaller values denote stronger invariance.

| | CMNIST | MetaShift |
|---|---|---|
| ERM | 12.0486 | 1.8824 |
| Vanilla mixup | 0.2769 | 0.2659 |
| IRM | 0.0112 | 0.8748 |
| DomainMix | 0.1674 | 1.1158 |
| **LISA (ours)** | **0.0012** | **0.2387** |

## 5 Theoretical Analysis

In this section, we provide some theoretical understandings that explain several of the empirical phenomena from the previous experiments and theoretically compare the worst-group errors of three methods: the proposed LISA, ERM, and vanilla mixup. Specifically, we consider a Gaussian mixture model with subpopulation and domain shifts, which has been widely adopted in theory to shed light upon complex machine learning phenomenon such as in Montanari et al. (2019); Zhang et al. (2021c). We also note here that despite the popularity of mixup in practice, the theoretical analysis of how mixup (with or without the selection strategies) affects the misclassification error is still largely unexplored in the literature even in the simple models. As discussed in Section 2, here, we define $y \in \{0, 1\}$ as the label, and $d \in \{B, G\}$ as the domain information. For $y \in \{0, 1\}$ and

$d \in \{B, G\}$, we consider the following model:

$$x_i | y_i = y, d_i = d \sim N(\mu^{(y,d)}, \Sigma^{(d)}), i = 1, \ldots, n^{(y,d)}, \tag{4}$$

where $\mu^{(y,d)} \in \mathbb{R}^p$ is the conditional mean vector and $\Sigma^{(d)} \in \mathbb{R}^{p \times p}$ is the covariance matrix. Let $n = \sum_{y \in \{0,1\}, d \in \{B,G\}} n^{(y,d)}$. Let $\pi^{(y,d)} = \mathbb{P}(y_i = y, d_i = d)$, $\pi^{(y)} = \mathbb{P}(y_i = y)$, and $\pi^{(d)} = \mathbb{P}(d_i = d)$.

To account for the spurious correlation brought by domains, we consider $\mu^{(y,B)} \neq \mu^{(y,G)}$ in general for $y \in \{0, 1\}$ and the imbalanced case where $\pi^{(0,B)}, \pi^{(1,G)} < 1/4$. Moreover, we assume there exists some invariance across different domains. Specifically, we assume

$$\mu^{(1,B)} - \mu^{(0,B)} = \mu^{(1,G)} - \mu^{(0,G)} := \Delta \text{ and } \Sigma^{(G)} = \Sigma^{(B)}.$$

According to the theory of Fisher's linear discriminant analysis rule (Anderson, 1962), the optimal classification rule is linear with slope $\Sigma^{-1}\Delta$. The assumption above implies that $(\Sigma^{-1}\Delta)^\top x$ is the (unknown) invariant representation in model equation 4.

Suppose we use some method $A$ and obtain a linear classifier $x^T b + b_0 > 0$ from a training data, we will apply it to a test data and compute the worst-group misclassification error, where the mis-classification error for domain $d$ and class $y$ is $E^{(y,d)}(b, b_0) := \mathbb{P}(\mathbb{1}(x_i^T b + b_0 > \frac{1}{2}) \neq y | d_i = d, y_i = y)$, and we denote the worst-group error with the method $A$ as

$$E_A^{(wst)} = \max_{d \in \{B,G\}, y \in \{0,1\}} E^{(y,d)}(b_A, b_{0,A}),$$

where $b_A$ and $b_{0,A}$ are the slope and intercept based on the method $A$. Specifically, $A = \text{ERM}$ denotes the ERM method (by minimizing the sum of squares loss on the training data altogether), $A = \text{mix}$ denotes the vanilla mixup method (without any selection strategy), and $A = \text{LISA}$ denotes the mixup strategy for LISA. We also denote its finite sample version by $\hat{E}_A^{(wst)}$.

Let $\widetilde{\Delta} = \mathbb{E}[x_i | y_i = 1] - \mathbb{E}[x_i | y_i = 0]$ denote the marginal difference and $\xi = \frac{\Delta^T \Sigma^{-1} \widetilde{\Delta}}{\|\Delta\|_\Sigma \|\widetilde{\Delta}\|_\Sigma}$ denote the correlation operator between the domain-specific difference $\Delta$ and the marginal difference $\widetilde{\Delta}$ with respect to $\Sigma$. We see that smaller $\xi$ indicates larger discrepancy between the marginal difference and the domain-specific difference and therefore implies stronger spurious correlation between the domains and labels. We present the following theorem showing that our proposed LISA algorithm outperforms the ERM and vanilla mixup in the subpopulation shifts setting.

**Theorem 1** (Worst-group error comparison with subpopulation shifts). *Consider $n$ independent samples generated from model (4), $\pi^{(B)} = \pi^{(1)} = 1/2$, $\pi^{(0,B)} = \pi^{(1,G)} = \alpha < 1/4$, $\max_{y,d} \|\mu^{(y,d)}\|_2 \leq C$, and $\Sigma$ is positive definite. Suppose $(\xi, \alpha)$ satisfies that $\xi \leq \min\{\frac{\|\widetilde{\Delta}\|_\Sigma}{\|\Delta\|_\Sigma}, 1\} - C\alpha$ for some large enough constant $C$ and $\|\widetilde{\Delta}\|_\Sigma \leq \sqrt{\frac{2\mathbb{E}[\lambda_i^2]}{\max\{3var(\lambda_i), 1/4\}}}$. Then for any $p_{sel} \in [0, 1]$,*

$$\hat{E}_{\text{LISA}}^{(wst)} \leq \min\{\hat{E}_{\text{ERM}}^{(wst)}, \hat{E}_{\text{mix}}^{(wst)}\} + O_P\left(\frac{p \log n}{n} + \frac{p}{\alpha n}\right),$$

where $p$ is the dimension of $x$. Theorem 1 implies that when $\xi$ is small (indicating that the domain has strong spurious correlation with the label) and $p = o(\alpha n)$, the worst-group classification errors of LISA are asymptotically smaller than that of ERM and vanilla mixup. In fact, our analysis shows that LISA yields a classification rule closer to the invariant classification rules by leveraging the domain information.

In the next theorem, we present the mis-classification error comparisons with domain shifts. That is, consider samples from a new unseen domain:

$$x_i^{(0,*)} \sim N(\mu^{(0,*)}, \Sigma), \quad x_i^{(1,*)} \sim N(\mu^{(1,*)}, \Sigma).$$

Let $\widetilde{\Delta}^* = 2(\mu^{(0,*)} - \mathbb{E}[x_i])$, where $\mathbb{E}[x_i]$ is the mean of the training distribution, and assume $\mu^{(1,*)} - \mu^{(0,*)} = \Delta$. Let $\xi^* = \frac{\widetilde{\Delta}^T \Sigma^{-1} \widetilde{\Delta}^*}{\|\widetilde{\Delta}\|_\Sigma \|\Delta\|_\Sigma}$ and $\gamma = \frac{\Delta^T \Sigma^{-1} \widetilde{\Delta}^*}{\|\widetilde{\Delta}\|_\Sigma \|\Delta\|_\Sigma}$ denote the correlation for $(\widetilde{\Delta}^*, \widetilde{\Delta})$ and for $(\widetilde{\Delta}^*, \Delta)$, respectively, with respect to $\Sigma^{-1}$. Let $E_A^{(wst*)} = \max_{y \in \{0,1\}} E^{(y,*)}(b_A, b_{0,A})$ and its sample version be $\hat{E}_A^{(wst*)}$.

**Theorem 2** (Mis-classification error comparison with domain shifts). *Suppose $n$ samples are independently generated from model (4), $\pi^{(B)} = \pi^{(1)} = 1/2, \pi^{(0,B)} = \pi^{(1,G)} = \alpha < 1/4$, $\max_{y,d} \|\mu^{(y,d)}\|_2 \leq C$ and $\Sigma$ is positive definite. Suppose that $(\xi, \xi^*, \gamma)$ satisfy that $0 \leq \xi^* \leq \gamma\xi$ and $\xi \leq \min\{\frac{\gamma}{2}\frac{\|\widetilde{\Delta}\|_\Sigma}{\|\Delta\|_\Sigma}, 1\} - C\alpha$ for some large enough constant $C$ and $\|\widetilde{\Delta}\|_\Sigma \leq \sqrt{\frac{2\mathbb{E}[\lambda_i^2]}{\max\{3var(\lambda_i), 1/4\}}}$. Then for any $p_{sel} \in [0, 1]$,*

$$\widehat{E}_{\text{LISA}}^{(wst*)} \leq \min\{\widehat{E}_{\text{ERM}}^{(wst*)}, \widehat{E}_{\text{mix}}^{(wst*)}\} + O_P\left(\frac{p \log n}{n} + \frac{p}{\alpha n}\right).$$

Similar to Theorem 1, this result shows that when domain has strong spurious correlation with the label (corresponding to small $\xi$), such a spurious correlation leads to the downgraded performance of ERM and vanilla mixup, while our proposed LISA method is able to mitigate such an issue by selective data interpolation. Proofs of Theorem 1 and Theorem 2 are provided in Appendix B.

## 6  RELATED WORK AND DISCUSSION

In this paper, we focus on improving the robustness of machine learning models to domain shifts and subpopulation shifts. Here, we discuss related approaches from the following three categories:

**Learning Invariant Representations with Domain Alignment.** Motivated by unsupervised domain adaptation (Ben-David et al., 2010; Ganin et al., 2016), the first category of works learns invariant representations by aligning representations across domains. The major research line of this category aims to eliminate the domain dependency by minimizing the divergence of feature distributions with different distance metrics, e.g., maximum mean discrepancy (Tzeng et al., 2014; Long et al., 2015), an adversarial loss (Ganin et al., 2016; Li et al., 2018), Wassertein distance (Zhou et al., 2020a). Follow-up works applied data augmentation to (1) generate more domains and enhance the consistency of representations during training (Yue et al., 2019; Zhou et al., 2020b; Xu et al., 2020; Yan et al., 2020; Shu et al., 2021; Wang et al., 2020) or (2) generate new domains in an adversarial way to imitate the challenging domains without using training domain information (Zhao et al., 2020; Qiao et al., 2020; Volpi et al., 2018). Unlike these latter methods, LISA instead focuses on canceling out correlations in the dataset between the domain and the label through selective data interpolation, leading to stronger empirical performance.

**Learning Invariant Representations with Invariant Predictors.** Beyond using domain alignment to learning invariant representations, recent work aims to further enhance the correlations between the invariant representations and the labels (Koyama & Yamaguchi, 2020). Representatively, motivated by casual inference, invariant risk minimization (IRM) (Arjovsky et al., 2019) aims to find a predictor that performs well across all domains. After IRM, the following works propose stronger regularizers by penalizing the variance of risks across all domains (Krueger et al., 2021), by aligning the gradient across domains (Koyama & Yamaguchi, 2020), by smoothing the cross-domain interpolation paths (Chuang & Mroueh, 2021), or through game-theoretic invariant rationalization criterion (Chang et al., 2020). Instead of using regularization, LISA eliminates spurious correlations in the data directly via data interpolation.

**Group Robustness.** The last category of methods combats spurious correlations and are particularly suitable for subpopulation shifts. These approaches include directly optimizing the worst-group performance with Distributionally Robust Optimization (Sagawa et al., 2020a; Zhang et al., 2021a; Zhou et al., 2021), generating samples around the minority groups (Goel et al., 2021), and balancing the majority and minority groups via reweighting (Sagawa et al., 2020b) or regularizing (Cao et al., 2019; 2020). Here, LISA proposes a more general strategy based on data augmentation that is suitable for both domain shifts and subpopulation shifts.

## 7  CONCLUSION

To tackle the distribution shifts, we propose LISA, a simple and efficient algorithm, to improve the out-of-distribution robustness. LISA aims to eliminate the domain-related spurious correlations among the training set by selective sample interpolation. We evaluate the effectiveness of LISA on nine datasets under subpopulation shifts and domain shifts settings, demonstrating its promise. Our theoretical results further strengthen the superiority of LISA by showing smaller worst-group mis-classification error compared with ERM and vanilla mixup.

## REPRODUCIBILITY STATEMENT

We conduct experiments under the setting of domain shifts and subpopulation shifts problems. In terms of the domain shifts, the results are reported in in Table 1. The dataset details are provided in Appendix A.1.1, and the training details along with the hyperparameter settings are in appendix A.1.2. Then, for the subpopulation shifts, we have included the full results including the error bounds in Table 16. The dataset details are discussed in Appendix A.2.1, while the training details and the hyperparameter settings are in Appendix A.2.2. We will release the code upon publication. Besides, The detailed proof of Theorem 1 and Theorem 2 are provided in Appendix B.

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

# A  ADDITIONAL EXPERIMENTS

## A.1  DOMAIN SHIFTS

### A.1.1  DATASET DETAILS

In this section, we provide detailed descriptions of datasets used in the experiments of domain shifts and report the data statistics in Table 8.

**Camelyon17**  We use Camelyon17 from the WILDS benchmark (Koh et al., 2021; Bandi et al., 2018), which provides $450,000$ lymph-node scans sampled from 5 hospitals. Camelyon17 is a medical image classification task where the input $x$ is a $96 \times 96$ image and the label $y$ is whether there exists tumor tissue in the image. The domain $d$ denotes the hospital that the patch was taken from. The training dataset is drawn from the first 3 hospitals, while out-of-distribution validation and out-of-distribution test datasets are sampled from the 4-th hospital and 5-th hospital respectively.

**FMoW**  The FMoW dataset is from the WILDS benchmark (Koh et al., 2021; Christie et al., 2018) — a satellite image classification task which includes 62 classes and 80 domains (16 years x 5 regions). Concretely, the input $x$ is a $224 \times 224$ RGB satellite image, the label $y$ is one of the 62 building or land use categories, and the domain $d$ represents the year that the image was taken as well as its corresponding geographical region – Africa, the Americas, Oceania, Asia, or Europe. The train/test/validation splits are based on the time when the images are taken. Specifically, images taken before 2013 are used as the training set. Images taken between 2013 and 2015 are used as the validation set. Images taken after 2015 are used for testing.

**RxRx1**  RxRx1 (Koh et al., 2021; Taylor et al., 2019) from the WILDS benchmark is a cell image classification task. In the dataset, some cells have been genetically perturbed by siRNA. The goal of RxRx1 is to predict which siRNA that the cells have been treated with. Concretely, the input $x$ is an image of cells obtained by fluorescent microscopy, the label $y$ indicates which of the $1,139$ genetic treatments the cells received, and the domain $d$ denotes the experimental batches. Here, 33 different batches of images are used for training, where each batch contains one sample for each class. The out-of-distribution validation set has images from 4 experimental batches. The out-of-distribution test set has 14 experimental batches. The average accuracy on out-of-distribution test set is reported.

**Amazon**  Each task in the Amazon benchmark (Koh et al., 2021; Ni et al., 2019) is a multi-class sentiment classification task. The input $x$ is the text of a review, the label $y$ is the corresponding star rating ranging from 1 to 5, and the domain $d$ is the corresponding reviewer. The training set has $245,502$ reviews from $1,252$ reviewers, while the out-of-distribution validation set has $100,050$ reviews from another $1,334$ reviewers. The out-of-distribution test set also has $100,050$ reviews from the rest $1,252$ reviewers. We evaluate the models by the 10th percentile of per-user accuracies in the test set.

**MetaShift**  We use the MetaShift (Liang & Zou, 2021), which is derived from Visual Genome (Krishna et al., 2016). MetaShift leverages the natural heterogeneity of Visual Genome to provide many distinct data distributions for a given class (e.g. "cats with cars" or "cats in bathroom" for the "cat" class). A key feature of MetaShift is that it provides explicit explanations of the dataset correlation and a distance score to measure the degree of distribution shift between any pair of sets. We adopt the "Cat vs. Dog" task in MetaShift, where we evaluate the model on the "dog(*shelf*)" domain with 306 images, and the "cat(*shelf*)" domain with 235 images. The training data for the "Cat" class is the cat(*sofa* + *bed*), including cat(*sofa*) domain and cat(*bed*) domain. MetaShift provides 4 different sets of training data for the "Dog" class in an increasingly challenging order, i.e., increasing the amount of distribution shift. Specifically, dog(*cabinet* + *bed*), dog(*bag* + *box*), dog(*bench* + *bike*), dog(*boat* + *surfboard*) are selected for training, where their corresponding distances to dog(*shelf*) are 0.44, 0.71, 1.12, 1.43.

Table 8: Dataset Statistics for Domain Shifts.

| Datasets | Domains | Metric | Base Model | Num. of classes |
|---|---|---|---|---|
| Camelyon17 | 5 hospitals | Avg. Acc. | DenseNet-121 | 2 |
| FMoW | 16 years x 5 regions | Worst-group Acc. | DenseNet-121 | 62 |
| RxRx1 | 51 experimental batches | Avg. Acc. | ResNet-50 | 1,139 |
| Amazon | 7,676 reviewers | 10th Percentile Acc. | DistilBERT-uncased | 5 |
| MetaShift | 4 backgrounds | Worst-group Acc. | ResNet-50 | 2 |

### A.1.2  TRAINING DETAILS

Follow WILDS Koh et al. (2021), we adopt pre-trained DenseNet121 (Huang et al., 2017) for Camelyon17 and FMoW datasets, ResNet-50 (He et al., 2016) for RxRx1 and MetaShift datasets, and DistilBert (Sanh et al., 2019) for Amazon datasets.

In each training iteration, we first draw a batch of samples $B_1$ from the training set. With $B_1$, we then select another sample batch $B_2$ with same labels as $B_1$ for data interpolation. The interpolation ratio $\lambda$ is drawn from the distribution $\text{Beta}(2, 2)$. We use the same image transformers as Koh et al. (2021), and all other hyperparameter settings are listed in Table 9.

Table 9: Hyperparameter settings for the domain shifts.

| Dataset | Camelyon17 | FMoW | RxRx1 | Amazon | MetaShift |
|---|---|---|---|---|---|
| Learning rate | 1e-4 | 1e-4 | 1e-3 | 2e-6 | 1e-3 |
| Weight decay | 0 | 0 | 1e-5 | 0 | 1e-4 |
| Scheduler | n/a | n/a | Cosine Warmup | n/a | n/a |
| Batch size | 32 | 32 | 72 | 8 | 16 |
| Type of mixup | CutMix | CutMix | CutMix | ManifoldMix | CutMix |
| Architecture | DenseNet121 | DenseNet121 | ResNet50 | DistilBert | ResNet50 |
| Optimizer | SGD | Adam | Adam | Adam | SGD |
| Maximum Epoch | 2 | 5 | 90 | 3 | 100 |
| Strategy sel. prob. $p_{sel}$ | 1.0 | 1.0 | 1.0 | 1.0 | 1.0 |

### A.1.3  FULL RESULTS OF WILDS DATA

Follow Koh et al. (2021), we reported more results on WILDS datasets in Table 10 - Table 13, including validation performance and the results of other metrics. According to these additional results, we could see that LISA outperforms other baseline approaches in all scenarios. Particularly, we here discuss two additional findings: (1) In Camelyon dataset, the test data is much more visually distinctive compared with the validation data, resulting in the large gap ($\sim 10\%$) between validation and test performance of ERM (see Table 10). However, LISA significantly reduces the performance gap between the validation and test sets, showing its promise in improving OOD robustness; (2) In Amazon dataset, though LISA performs worse than ERM in average accuracy, it achieves the best accuracy at the 10th percentile, which is regarded as a more common and important metric to evaluate whether models perform consistently well across all users (Koh et al., 2021).

Table 10: Full Results of Camelyon17. We report both validation accuracy and test accuracy.

|  | Validation Acc. | Test Acc. |
|---|---|---|
| ERM | $84.9 \pm 3.1\%$ | $70.3 \pm 6.4\%$ |
| IRM | $\mathbf{86.2 \pm 1.4\%}$ | $64.2 \pm 8.1\%$ |
| Coral | $\mathbf{86.2 \pm 1.4\%}$ | $59.5 \pm 7.7\%$ |
| GroupDRO | $85.5 \pm 2.2\%$ | $68.4 \pm 7.3\%$ |
| DomainMix | $83.5 \pm 1.1\%$ | $69.7 \pm 5.5\%$ |
| Fish | $83.9 \pm 1.2\%$ | $74.7 \pm 7.1\%$ |
| **LISA (ours)** | $81.8 \pm 1.3\%$ | $\mathbf{77.1 \pm 6.5\%}$ |

Table 11: Full Results of FMoW. Here, we report the average accuracy and the worst-domain accuracy on both validation and test sets.

|  | Validation | | Test | |
|---|---|---|---|---|
|  | Avg. Acc. | Worst Acc. | Avg. Acc. | Worst Acc. |
| ERM | $\mathbf{59.5 \pm 0.37\%}$ | $48.9 \pm 0.62\%$ | $\mathbf{53.0 \pm 0.55\%}$ | $32.3 \pm 1.25\%$ |
| IRM | $57.4 \pm 0.37\%$ | $47.5 \pm 1.57\%$ | $50.8 \pm 0.13\%$ | $30.0 \pm 1.37\%$ |
| Coral | $56.9 \pm 0.25\%$ | $47.1 \pm 0.43\%$ | $50.5 \pm 0.36\%$ | $31.7 \pm 1.24\%$ |
| GroupDRO | $58.8 \pm 0.19\%$ | $46.5 \pm 0.25\%$ | $52.1 \pm 0.50\%$ | $30.8 \pm 0.81\%$ |
| DomainMix | $58.6 \pm 0.29\%$ | $48.9 \pm 1.15\%$ | $51.6 \pm 0.19\%$ | $34.2 \pm 0.76\%$ |
| Fish | $57.8 \pm 0.15\%$ | $\mathbf{49.5 \pm 2.34\%}$ | $51.8 \pm 0.32\%$ | $34.6 \pm 0.18\%$ |
| **LISA (ours)** | $58.7 \pm 0.92\%$ | $48.7 \pm 0.74\%$ | $\mathbf{52.8 \pm 0.94\%}$ | $\mathbf{35.5 \pm 0.65\%}$ |

## A.2 SUBPOPULATION SHIFTS

### A.2.1 DATASET DETAILS

We detail the data descriptions of subpopulation shifts below and report the detailed data statistics in Table 14.

**Colored MNIST (CMNIST)**: We classify MNIST digits from 2 classes, where classes 0 and 1 indicate original digits (0,1,2,3,4) and (5,6,7,8,9). The color is treated as a spurious attribute. Concretely, in the training set, the proportion between red samples and green samples is 8:2 in class 0, while the proportion is set as 2:8 in class 1. In the validation set, the proportion between green and red samples is 1:1 for all classes. In the test set, the proportion between green and red samples is 1:9 in class 0, while the ratio is 9:1 in class 1. The data sizes of train, validation, and test sets are 30000, 10000, and 20000, respectively.

**Waterbirds** (Sagawa et al., 2020a): The Waterbirds dataset aims to classify birds as "waterbird" or "landbird", where each bird image is spuriously associated with the background "water" or "land". Waterbirds is a synthetic dataset where each image is composed by pasting a bird image sampled from CUB dataset (Wah et al., 2011) to a background drawn from the Places dataset Zhou et al. (2017). The bird categories in CUB are stratified as land birds or water birds. Specifically, the following bird species are selected to construct the waterbird class: albatross, auklet, cormorant, frigatebird, fulmar, gull, jaeger, kittiwake, pelican, puffin, tern, gadwall, grebe, mallard, merganser, guillemot, or Pacific loon. All other bird species are combined as the landbird class. We define (land background, waterbird) and (water background, landbird) are minority groups. There are 4,795 training samples while only 56 samples are "waterbirds on land" and 184 samples are "landbirds on water". The remaining training data include 3,498 samples from "landbirds on land", and 1,057 samples from "waterbirds on water".

**CelebA** (Liu et al., 2015; Sagawa et al., 2020a): For the CelebA data (Liu et al., 2015), we follow the data preprocess procedure from Sagawa et al. (2020a). CelebA defines a image classification task where the input is a face image of celebrities and the classification label is its corresponding hair color – "blond" or "not blond." The label is spuriously correlated with gender, i.e., male or female. In CelebA, the minority groups are (blond, male) and (not blond, female). The number of

Table 12: Full Results of RxRx1. ID: in-distribution; OOD: out-of-distribution

|  | Validation Acc. | Test ID Acc. | Test OOD Acc. |
|---|---|---|---|
| ERM | $19.4 \pm 0.2\%$ | $35.9 \pm 0.4\%$ | $29.9 \pm 0.4\%$ |
| IRM | $5.6 \pm 0.4\%$ | $9.9 \pm 1.4\%$ | $8.2 \pm 1.1\%$ |
| Coral | $18.5 \pm 0.4\%$ | $34.0 \pm 0.3\%$ | $28.4 \pm 0.3\%$ |
| GroupDRO | $15.2 \pm 0.1\%$ | $28.1 \pm 0.3\%$ | $23.0 \pm 0.3\%$ |
| DomainMix | $19.3 \pm 0.7\%$ | $39.8 \pm 0.2\%$ | $30.8 \pm 0.4\%$ |
| Fish | $7.5 \pm 0.6\%$ | $12.7 \pm 1.9\%$ | $10.1 \pm 1.5\%$ |
| **LISA (ours)** | $\mathbf{20.1 \pm 0.4\%}$ | $\mathbf{41.2 \pm 1.0\%}$ | $\mathbf{31.9 \pm 0.8\%}$ |

Table 13: Full Results of Amazon. Both the average accuracy and the 10th Percentile accuracy are reported.

|  | Validation | | Test | |
|---|---|---|---|---|
|  | Avg. Acc. | 10-th Per. | Avg. Acc. | 10-th Per. Acc. |
| ERM | $\mathbf{72.7 \pm 0.1\%}$ | $\mathbf{55.2 \pm 0.7\%}$ | $\mathbf{71.9 \pm 0.1\%}$ | $53.8 \pm 0.8\%$ |
| IRM | $71.5 \pm 0.3\%$ | $54.2 \pm 0.8\%$ | $70.5 \pm 0.3\%$ | $52.4 \pm 0.8\%$ |
| Coral | $72.0 \pm 0.3\%$ | $54.7 \pm 0.0\%$ | $70.0 \pm 0.6\%$ | $52.9 \pm 0.8\%$ |
| GroupDRO | $70.7 \pm 0.6\%$ | $54.7 \pm 0.0\%$ | $70.0 \pm 0.6\%$ | $53.3 \pm 0.0\%$ |
| DomainMix | $71.9 \pm 0.2\%$ | $54.7 \pm 0.0\%$ | $71.1 \pm 0.1\%$ | $53.3 \pm 0.0\%$ |
| Fish | $72.5 \pm 0.0\%$ | $54.7 \pm 0.0\%$ | $71.7 \pm 0.1\%$ | $53.3 \pm 0.0\%$ |
| **LISA (ours)** | $71.2 \pm 0.3\%$ | $\mathbf{55.1 \pm 0.61\%}$ | $70.6 \pm 0.3\%$ | $\mathbf{54.7 \pm 0.0\%}$ |

samples for each group are 71,629 "dark hair, female", 66,874 "dark hair, male", 22,880 "blond hair, female", 1,387 "blond hair, male".

**CivilComments** (Borkan et al., 2019; Koh et al., 2021): We use CivilComments from the WILDS benchmark (Koh et al., 2021). CivilComments is a text classification task, aiming to predict whether an online comment is toxic or non-toxic. The spurious domain identifications are defined as the demographic features, including male, female, LGBTQ, Christian, Muslim, other religion, Black, and White. CivilComments contains 450,000 comments collected from online articles. The number of samples for training, validation, and test are 269,038, 45,180, and 133,782, respectively. The readers may kindly refer to Table 17 in Koh et al. (2021) for the detailed group information.

Table 14: Dataset Statistics for Subpopulation Shifts. All datasets are binary classification tasks and we use the worst group accuracy as the evaluation metric.

| Datasets | Domains | Base Model | Class Information |
|---|---|---|---|
| CMNIST | 2 digit colors | ResNet-50 | digit (0,1,2,3,4) v.s. (5,6,7,8,9) |
| Waterbirds | 2 backgrounds | ResNet-50 | waterbirds v.s. landbirds |
| CelebA | 2 hair colors | ResNet-50 | man v.s. women |
| CivilComments | 8 demographic identities | DistilBERT-uncased | toxic v.s. non-toxic |

### A.2.2 Training Details

We adopt pre-trained ResNet-50 (He et al., 2016) and BERT (Sanh et al., 2019) as the model for image data (i.e., CMNIST, Waterbirds, CelebA) and text data (i.e., CivilComments), respectively. In each training iteration, we sample a batch of data per group. For sample selection strategy I, we randomly apply mixup on sample batches with the same labels but different domains. For sample selection strategy II, we instead apply mixup on sample batches with the same domain but different labels. The interpolation ratio $\lambda$ is sampled from the distribution $\text{Beta}(2, 2)$. All hyperparameters are listed in Table 15.

Table 15: Hyperparameter settings for the subpopulation shifts.

| Dataset | CMNIST | Waterbirds | CelebA | CivilComments |
|---|---|---|---|---|
| Learning rate | 1e-3 | 1e-3 | 1e-4 | 1e-5 |
| Weight decay | 1e-4 | 1e-4 | 1e-4 | 0 |
| Scheduler | n/a | n/a | n/a | n/a |
| Batch size | 16 | 16 | 16 | 8 |
| Type of mixup | mixup | mixup | CutMix | ManifoldMix |
| Architecture | ResNet50 | ResNet50 | ResNet50 | DistilBert |
| Optimizer | SGD | SGD | SGD | Adam |
| Maximum Epoch | 300 | 300 | 50 | 3 |
| Strategy sel. prob. $p_{sel}$ | 0.5 | 0.5 | 0.5 | 1.0 |

### A.2.3 ADDITIONAL RESULTS

In this section, we have added the full results of subpopulation shifts in Table 16 and Table 17.

Table 16: Full results of subpopulation shifts with standard deviation. All the results are performed with three random seed.

| | CMNIST | | Waterbirds | |
|---|---|---|---|---|
| | Avg. | Worst | Avg. | Worst |
| ERM | $27.8 \pm 1.9\%$ | $0.0 \pm 0.0\%$ | $97.0 \pm 0.2\%$ | $63.7 \pm 1.9\%$ |
| UW | $72.2 \pm 1.1\%$ | $66.0 \pm 0.7\%$ | $95.1 \pm 0.3\%$ | $88.0 \pm 1.3\%$ |
| IRM | $72.1 \pm 1.2\%$ | $70.3 \pm 0.8\%$ | $87.5 \pm 0.7\%$ | $75.6 \pm 3.1\%$ |
| Coral | $71.8 \pm 1.7\%$ | $69.5 \pm 0.9\%$ | $90.3 \pm 1.1\%$ | $79.8 \pm 1.8\%$ |
| GroupDRO | $72.3 \pm 1.2\%$ | $68.6 \pm 0.8\%$ | $91.8 \pm 0.3\%$ | $\mathbf{90.6 \pm 1.1\%}$ |
| DomainMix | $51.4 \pm 1.3\%$ | $48.0 \pm 1.3\%$ | $76.4 \pm 0.3\%$ | $53.0 \pm 1.3\%$ |
| Fish | $46.9 \pm 1.4\%$ | $35.6 \pm 1.7\%$ | $85.6 \pm 0.4\%$ | $64.0 \pm 0.3\%$ |
| **LISA** | $74.0 \pm 0.1\%$ | $\mathbf{73.3 \pm 0.2\%}$ | $91.8 \pm 0.3\%$ | $89.2 \pm 0.6\%$ |

| | CelebA | | CivilComments | |
|---|---|---|---|---|
| | Avg. | Worst | Avg. | Worst |
| ERM | $94.9 \pm 0.2\%$ | $47.8 \pm 3.7\%$ | $92.2 \pm 0.1\%$ | $56.0 \pm 3.6\%$ |
| UW | $92.9 \pm 0.2\%$ | $83.3 \pm 2.8\%$ | $89.8 \pm 0.5\%$ | $69.2 \pm 0.9\%$ |
| IRM | $94.0 \pm 0.4\%$ | $77.8 \pm 3.9\%$ | $88.8 \pm 0.7\%$ | $66.3 \pm 2.1\%$ |
| Coral | $93.8 \pm 0.3\%$ | $76.9 \pm 3.6\%$ | $88.7 \pm 0.5\%$ | $65.6 \pm 1.3\%$ |
| GroupDRO | $92.1 \pm 0.4\%$ | $87.2 \pm 1.6\%$ | $89.9 \pm 0.5\%$ | $70.0 \pm 2.0\%$ |
| DomainMix | $93.4 \pm 0.1\%$ | $65.6 \pm 1.7\%$ | $90.9 \pm 0.4\%$ | $63.6 \pm 2.5\%$ |
| Fish | $93.1 \pm 0.3\%$ | $61.2 \pm 2.5\%$ | $89.8 \pm 0.4\%$ | $71.1 \pm 0.4\%$ |
| **LISA (ours)** | $92.4 \pm 0.4\%$ | $\mathbf{89.3 \pm 1.1\%}$ | $89.2 \pm 0.9\%$ | $\mathbf{72.6 \pm 0.1\%}$ |

### A.3 RESULTS ON DATASETS WITHOUT SPURIOUS CORRELATIONS

In order to analyze the factors that lead to the performance gains of LISA, we conduct experiments on datasets without spurious correlations. To be more specific, we balance the number of samples for each group under the subpopulation shifts setting. The results of ERM, Vanilla mixup and LISA on CMNIST, Waterbirds and CelebA are reported in Table 18. The results show that LISA performs similarly compared with ERM when datasets do not have spurious correlations. If there exists any spurious correlation, LISA significantly outperforms ERM. Another interesting finding is that Vanilla mixup outperforms LISA and ERM without spurious correlations, while LISA achieves the best performance with spurious correlations. This finding strengthens our conclusion that the performance gains of LISA are caused by eliminating spurious correlations rather than data augmentation.

Table 17: Full table of the comparison between LISA and other substitute mixup strategies in sub-population shifts. UW represents upweighting.

| | CMNIST | | Waterbirds | |
| | Avg. | Worst | Avg. | Worst |
| --- | --- | --- | --- | --- |
| ERM | $27.8 \pm 1.9\%$ | $0.0 \pm 0.0\%$ | $97.0 \pm 0.2\%$ | $63.7 \pm 1.9\%$ |
| Vanilla mixup | $32.6 \pm 3.1\%$ | $3.1 \pm 2.4\%$ | $81.0 \pm 0.2\%$ | $56.2 \pm 0.2\%$ |
| Vanilla mixup + UW | $72.2 \pm 0.7\%$ | $71.8 \pm 0.1\%$ | $92.1 \pm 0.1\%$ | $85.6 \pm 1.0\%$ |
| In-group Group | $33.6 \pm 1.9\%$ | $24.0 \pm 1.1\%$ | $88.7 \pm 0.3\%$ | $68.0 \pm 0.4\%$ |
| In-group + UW | $72.6 \pm 0.1\%$ | $71.6 \pm 0.2\%$ | $91.4 \pm 0.6\%$ | $87.1 \pm 0.6\%$ |
| **LISA (ours)** | $74.0 \pm 0.1\%$ | $\mathbf{73.3 \pm 0.2\%}$ | $91.8 \pm 0.3\%$ | $\mathbf{89.2 \pm 0.6\%}$ |
| | CelebA | | CivilComments | |
| | Avg. | Worst | Avg. | Worst |
| ERM | $94.9 \pm 0.2\%$ | $47.8 \pm 3.7\%$ | $92.2 \pm 0.1\%$ | $56.0 \pm 3.6\%$ |
| Vanilla mixup | $95.8 \pm 0.0\%$ | $46.4 \pm 0.5\%$ | $90.8 \pm 0.8\%$ | $67.2 \pm 1.2\%$ |
| Vanilla mixup + UW | $91.5 \pm 0.2\%$ | $88.0 \pm 0.3\%$ | $87.8 \pm 1.2\%$ | $66.1 \pm 1.4\%$ |
| Within Group | $95.2 \pm 0.3\%$ | $58.3 \pm 0.9\%$ | $90.8 \pm 0.6\%$ | $69.2 \pm 0.8\%$ |
| Within Group + UW | $92.4 \pm 0.4\%$ | $87.8 \pm 0.6\%$ | $84.8 \pm 0.7\%$ | $69.3 \pm 1.1\%$ |
| **LISA (ours)** | $92.4 \pm 0.4\%$ | $\mathbf{89.3 \pm 1.1\%}$ | $89.2 \pm 0.9\%$ | $\mathbf{72.6 \pm 0.1\%}$ |

Table 18: Results on Datasets without Spurious Correlations

| Dataset | ERM | Vanilla mixup | LISA |
| --- | --- | --- | --- |
| CMNIST | 73.67% | 74.28% | 73.18% |
| Waterbirds | 88.07% | 88.23% | 87.05% |
| CelebA | 86.11% | 88.89% | 87.22% |

# B PROOFS OF THEOREM 1 AND THEOREM 2

**Outline of the proof**. We will first find the mis-classification errors based on the population version of OLS with different mixup strategies. Next, we will develop the convergence rate of the empirical OLS based on $n$ samples towards its population version. These two steps together give us the empirical mis-classification errors of different methods. We will separately show that the upper bounds in Theorem 1 and Theorem 2 hold for two strategies of LISA and hence hold for any $p_{sel} \in [0, 1]$. Let L1 denote selection strategy I of LISA method and L2 denote selection strategy II of LISA method.

Let $\pi_1 = \mathbb{P}(y_i = 1)$ and $\pi_0 = \mathbb{P}(y_i = 0)$ denote the marginal class proportions in the training samples. Let $\pi_B = \mathbb{P}(d_i = B)$ and $\pi_G = \mathbb{P}(d_i = G)$ denote the marginal subpopulation proportions in the training samples. Let $\pi_{G|1} = \mathbb{P}(d_i = G|y_i = 1)$ and define $\pi_{G|0}$, $\pi_{B|1}$, and $\pi_{B|0}$ similarly.

We consider the setting where $\alpha := \pi^{(1,G)} = \pi^{(0,B)}$ is relatively small and $\pi^{(1)} = \pi^{(0)} = \pi^{(G)} = \pi^{(B)} = 1/2$.

## B.1 DECOMPOSING THE LOSS FUNCTION

Recall that $\Delta = \mu^{(1,G)} - \mu^{(0,G)} = \mu^{(1,B)} - \mu^{(0,B)}$. We further define $\widetilde{\Delta} = \mu^{(1)} - \mu^{(0)}$, $\theta^{(G)} = \mu^{(0,G)} - \mathbb{E}[x_i]$, and $\theta^{(B)} = \mu^{(0,B)} - \mathbb{E}[x_i]$.

For the mixup estimators, we will repeatedly use the fact that $\lambda_i$ has a symmetric distribution with support $[0, 1]$.

For ERM estimator based on $(X, y)$, where $b_0 = \frac{1}{2} - \mathbb{E}[x_i]^T b$, we have

$$
\begin{aligned}
(\mu^{(0,G)})^T b + b_0 &= (\mu^{(0,G)} - \mathbb{E}[x_i])^T b + \frac{1}{2} \\
&= (\theta^{(G)})^T b + \mathbb{E}[y_i] \\
(\mu^{(1,G)})^T b + b_0 &= (\mu^{(1,G)} - \mathbb{E}[x_i])^T b + \frac{1}{2} \\
&= \Delta^T b + (\theta^{(G)})^T b + \mathbb{E}[y_i],
\end{aligned}
$$

Notice that based on the estimator $b, b_0$

$$
E^{(1,d)}(b, b_0) = \Phi\left(\frac{-\Delta^T b - (\theta^{(d)})^T b}{\sqrt{b^T \Sigma b}}\right) \text{ and } E^{(0,d)}(b, b_0) = \Phi\left(\frac{(\theta^{(d)})^T b}{\sqrt{b^T \Sigma b}}\right).
$$

### B.2 Classification errors of four methods with infinite training samples

We first provide the limit of the classification errors when $n \to \infty$.

#### B.2.1 Baseline method: ERM

For the training data, it is easy to show that

$$
\begin{aligned}
var(x) &= \mathbb{E}[var(x|y)] + var(\mathbb{E}[x|y]) \\
&= \Sigma + \mathbb{E}[var(\mathbb{E}[x|y, D]|y)] + var((\mu^{(1)} - \mu^{(0)})y) \\
&= \Sigma + \mathbb{E}[var(\mu^{(0,B)} - \mu^{(0,G)})\mathbb{1}(D = B)|y)] + \widetilde{\Delta}^{\otimes 2}\pi^{(1)}\pi^{(0)} \\
&= \Sigma + \frac{1}{2}(\mu^{(0,B)} - \mu^{(0,G)})^{\otimes 2}(\pi_{B|1}\pi_{G|1} + \pi_{B|0}\pi_{G|0}) + \widetilde{\Delta}^{\otimes 2}\pi^{(1)}\pi^{(0)} \\
cov(x, y) &= cov(\mathbb{E}[x|y], y) \\
&= cov(\mu^{(0)} + \widetilde{\Delta}y, y) \\
&= cov(\widetilde{\Delta}y, y) = \widetilde{\Delta}\pi^{(1)}\pi^{(0)}
\end{aligned}
$$

For $a_0 = \frac{1}{2}(\pi_{B|1}\pi_{G|1} + \pi_{B|0}\pi_{G|0})$ and $\Delta_0 = \mu^{(0,G)} - \mu^{(0,B)}$, the ERM has slope and intercept being

$$
\begin{aligned}
b &= var(x)^{-1}cov(x, y) \\
&\propto (\Sigma + a_0\Delta_0^{\otimes 2})^{-1}\widetilde{\Delta} \\
&= \Sigma^{-1}\widetilde{\Delta} - \Sigma^{-1}\Delta_0 \cdot \frac{a_0\widetilde{\Delta}^T\Sigma^{-1}\Delta_0}{1 + a_0\Delta_0^T\Sigma^{-1}\Delta_0} \\
b_0 &= \mathbb{E}[y] - \mathbb{E}[x^T b].
\end{aligned}
$$

In the extreme case where $\pi_{0,B} = \pi_{1,G} = 0$, we have

$$
\widetilde{\Delta} = \mu^{(1,B)} - \mu^{(0,G)}, \theta^{(G)} = -\frac{1}{2}\widetilde{\Delta}, \theta^{(B)} = \frac{1}{2}\widetilde{\Delta} - \Delta, \text{ and } \Delta_0 = \Delta - \widetilde{\Delta}.
$$

Hence,

$$
E_0^{(wst)} = \max\left\{\Phi\left(\frac{(\frac{1}{2}\widetilde{\Delta} - \Delta)^T b}{\sqrt{b^T \Sigma b}}\right), \Phi\left(\frac{-\frac{1}{2}\widetilde{\Delta}^T b}{\sqrt{b^T \Sigma b}}\right)\right\}, \tag{5}
$$

where $b$ is computed via ERM.

#### B.2.2 Baseline method: Vanilla mixup

The vanilla mixup does not use the group information. Let $i_1$ be a random draw from $\{1, \ldots, n\}$. Let $i_2$ be a random draw from $\{1, \ldots, n\}$ independent of $i_1$. Let

$$
\tilde{y}_i = \lambda_i y_{i_1} + (1 - \lambda_i)y_{i_2}
$$

and
$$\tilde{x}_i = \lambda_i x_{i_1} + (1 - \lambda_i) x_{i_2}.$$

We can find that
$$\begin{aligned}
cov(\tilde{x}_i, \tilde{y}_i) &= cov(\lambda_i x_{i_1} + (1 - \lambda_i) x_{i_2}, \lambda_i y_{i_1} + (1 - \lambda_i) y_{i_2}) \\
&= cov(\lambda_i x_{i_1}, \lambda_i y_{i_1}) + cov((1 - \lambda_i) x_{i_2}, (1 - \lambda_i) y_{i_2}) \\
&= (\mathbb{E}[\lambda_i^2] + \mathbb{E}[(1 - \lambda_i)^2]) cov(x_i, y_i). \\
cov(\tilde{x}_i) &= (\mathbb{E}[\lambda_i^2] + \mathbb{E}[(1 - \lambda_i)^2]) cov(x_i).
\end{aligned}$$

Hence, the population-level slope is the same as the slope in the benchmark method. It is easy to show that the population-level intercept is also the same. Hence,
$$E_{\text{mix}}^{(wst)} = E_0^{(wst)}.$$

### B.3 LISA WITH SELECTION STRATEGY (LISA-II): MIXUP WITHIN EACH CLASS

Define
$$x_i^{(\lambda)} = \lambda_i x_{i_1}^{(y_i, G)} + (1 - \lambda_i) x_{i_2}^{(y_i, B)},$$

where $i_1$ is a random draw from $\{l : y_l = y_i, D_l = G\}$ and $i_2$ is a random draw from $\{l : y_l = y_i, D_l = B\}$. Then we perform OLS based on $(x_i^{(\lambda)}, y_i), i = 1, \dots, n$.

We can calculate that
$$\begin{aligned}
cov(x_i^{(\lambda)}, y_i) &= cov(\mathbb{E}[x_i^{(\lambda)} | y_i], y_i) = cov(\frac{1}{2} \mu^{(y_i, G)} + \frac{1}{2} \mu^{(y_i, B)}, y_i) \\
&= var(y_i) \Delta = \pi^{(1)} \pi^{(0)} \Delta \\
cov(x_i^{(\lambda)}) &= \mathbb{E}[cov(x_i^{(\lambda)} | y_i, \lambda_i)] + cov(\mathbb{E}[x_i^{(\lambda)} | y_i, \lambda_i]) \\
&= 2\mathbb{E}[\lambda_i^2] \Sigma + cov(\lambda_i(\mu^{(0,G)} - \mu^{(0,B)}) + \Delta y_i) \\
&= 2\mathbb{E}[\lambda_i^2] \Sigma + var(\lambda_i)(\mu^{(0,G)} - \mu^{(0,B)})^{\otimes 2} + \pi^{(1)} \pi^{(0)} \Delta^{\otimes 2}.
\end{aligned}$$

Notice that
$$\mathbb{E}[x_i^{(\lambda)}] = \frac{1}{4}(\mu^{(0,G)} + \mu^{(1,G)} + \mu^{(0,B)} + \mu^{(1,B)}) = \frac{1}{2}(\mu^{(0,G)} + \mu^{(1,B)}) = \mathbb{E}[x_i].$$

For $a_{\text{L1}} = var(\lambda_i)/(2\mathbb{E}[\lambda_i^2])$ and $\Delta_0 = \mu^{(0,G)} - \mu^{(0,B)}$, the OLS has slope and intercept being
$$\begin{aligned}
b &= var(x_i^{(\lambda)})^{-1} cov(x_i^{(\lambda)}, y) \\
&\propto (\Sigma + \frac{var(\lambda_i)}{2\mathbb{E}[\lambda_i^2]}(\mu^{(0,G)} - \mu^{(0,B)})^{\otimes 2})^{-1} \Delta \\
&\propto \Sigma^{-1} \Delta - \Sigma^{-1} \Delta_0 \cdot \frac{a_{\text{L1}} \Delta^T \Sigma^{-1} \Delta_0}{1 + a_{\text{L1}} \Delta_0^T \Sigma^{-1} \Delta_0} \\
b_0 &= \mathbb{E}[y] - \mathbb{E}[(x^{(\lambda)})^T b].
\end{aligned}$$

$$E_{\text{L1}}^{(wst)} = \max\{\Phi\left(\frac{(\frac{1}{2}\tilde{\Delta} - \Delta)^T b}{\sqrt{b^T \Sigma b}}\right), \Phi\left(\frac{-\frac{1}{2}\tilde{\Delta}^T b}{\sqrt{b^T \Sigma b}}\right)\}, \tag{6}$$

where $b$ is computed based on $(x_i^{(\lambda)}, y_i), i = 1, \dots, n$.

### B.4 LISA WITH SELECTION STRATEGY II (LISA-II): MIXUP WITHIN EACH DOMAIN

The interpolated sample can be written as
$$\begin{aligned}
(\tilde{y}_i, \tilde{x}_i) &= (\lambda_i, \lambda_i x_{i_1}^{(1,G)} + (1 - \lambda_i) x_{i_2}^{(0,G)}) \text{ if } d_i = G \\
(\tilde{y}_i, \tilde{x}_i) &= (\lambda_i, \lambda_i x_{i_1}^{(1,B)} + (1 - \lambda_i) x_{i_2}^{(0,B)}) \text{ if } d_i = B,
\end{aligned}$$

where $i_1$ is a random draw from $\{l : d_l = d_i, y_i = 1\}$ and $i_2$ is a random draw from $\{l : d_l = d_i, y_i = 0\}$.

We consider regress $\tilde{y}_i$ on $\tilde{x}_i$.

$$cov(\tilde{x}_i, \tilde{y}_i | d_i = G) = cov(\mathbb{E}[\tilde{x}_i | \tilde{y}_i, d_i = G], \tilde{y}_i | d_i = G) = var(\tilde{y}_i)(\mu^{(1,G)} - \mu^{(0,G)})$$
$$var(\tilde{x}_i | d_i = G) = \mathbb{E}[var(\tilde{x}_i |, \lambda_i, D_i = G) | d_i = G] + var(\mathbb{E}[\tilde{x}_i |, \lambda_i, d_i = G] | D_i = G]$$
$$= 2\mathbb{E}[\lambda_i^2]\Sigma + var(\lambda_i \mu^{(1,G)} + (1 - \lambda_i)\mu^{(0,G)} | d_i = G)$$
$$= 2\mathbb{E}[\lambda_i^2]\Sigma + var(\tilde{y}_i)\Delta^{\otimes 2}.$$

We further have

$$cov(\tilde{x}_i, \tilde{y}_i) = \mathbb{E}[cov(\tilde{x}_i, \tilde{y}_i | d_i)] + cov(\mathbb{E}[\tilde{x}_i | d_i], \mathbb{E}[\tilde{y}_i | d_i])$$
$$= cov(\tilde{x}_i^{(G)}, \tilde{y}_i^{(G)})\pi^{(G)} + cov(\tilde{x}_i^{(B)}, \tilde{y}_i^{(B)})\pi^{(B)}$$
$$= var(\tilde{y}_i)(\mu^{(1,G)} - \mu^{(0,G)})\pi^{(G)} + var(\tilde{y}_i)(\mu^{(1,B)} - \mu^{(0,B)})\pi^{(B)}$$
$$= var(\tilde{y}_i)\Delta.$$

Moreover,

$$var(\tilde{x}_i) = \mathbb{E}[var(\tilde{x}_i | d_i)] + var(\mathbb{E}[\tilde{x}_i | d_i])$$
$$= var(\tilde{x}_i^{(G)})\pi^{(G)} + var(\tilde{x}^{(B)})\pi^{(B)} + (\mathbb{E}[\tilde{x}^{(G)}] - \mathbb{E}[\tilde{x}^{(B)}])^{\otimes 2}\pi^{(G)}\pi^{(B)}$$
$$= 2\mathbb{E}[\lambda_i^2]\Sigma + var(\lambda_i)\Delta^{\otimes 2} + (\mu^{(0,G)} - \mu^{(0,B)})^{\otimes 2}\pi^{(G)}\pi^{(B)}.$$

Slope:

$$b = var(\tilde{x}_i)^{-1}cov(\tilde{x}_i, \tilde{y}_i)$$
$$\propto (\Sigma + a_{L2}\Delta_0^{\otimes 2})^{-1}\Delta$$
$$= \Sigma^{-1}\Delta - \Sigma^{-1}\Delta_0 \cdot \frac{a_{L2}(\Delta_0)^T\Sigma^{-1}\Delta}{1 + a_{L2}(\Delta_0)^T\Sigma^{-1}\Delta_0},$$

where $a_{L2} = \frac{\pi^{(B)}\pi^{(G)}}{2\mathbb{E}[\lambda_i^2]}$.

Moreover, $b_0 = \mathbb{E}[\tilde{y}_i] - \mathbb{E}[\tilde{x}_i]^T b = \frac{1}{2} - \mathbb{E}[\tilde{x}_i]^T b$. Notice that

$$\mathbb{E}[\tilde{x}_i] = \frac{1}{4}(\mu^{(0,G)} + \mu^{(1,G)} + \mu^{(0,B)} + \mu^{(1,B)})$$
$$= \frac{1}{4}(2\mu^{(0,G)} + \Delta + 2\mu^{(1,B)} - \Delta)$$
$$= \frac{1}{2}(\mu^{(0,G)} + \mu^{(1,B)}) = \mathbb{E}[x_i].$$

Hence,

$$E_{L2}^{(wst)} = \max\{\Phi\left(\frac{(\frac{1}{2}\widetilde{\Delta} - \Delta)^T b}{\sqrt{b^T \Sigma b}}\right), \Phi\left(\frac{-\frac{1}{2}\widetilde{\Delta}^T b}{\sqrt{b^T \Sigma b}}\right)\}, \tag{7}$$

where $b$ is computed based on $(\tilde{x}_i, \tilde{y}_i)$, $i = 1, \ldots, n$.

**Method comparison**. We only need to compare (5), (6), and (7).

For the ERM, $0 \leq a_0 \leq 2\alpha$ and

$$b = (1 + \frac{a_0\widetilde{\Delta}^T\Sigma^{-1}\Delta_0}{1 + a_0\Delta_0^T\Sigma^{-1}\Delta_0})\Sigma^{-1}\widetilde{\Delta} - \frac{a_0\widetilde{\Delta}^T\Sigma^{-1}\Delta_0}{1 + a_0\Delta_0^T\Sigma^{-1}\Delta_0}\Sigma^{-1}\Delta$$

$$\propto \Sigma^{-1}\widetilde{\Delta} - \frac{a_0\widetilde{\Delta}^T\Sigma^{-1}\Delta_0}{1 + a_0\Delta_0^T\Sigma^{-1}\Delta_0 + a_0\widetilde{\Delta}^T\Sigma^{-1}\Delta_0}\Sigma^{-1}\Delta$$

$$\propto \Sigma^{-1}\widetilde{\Delta} - \frac{a_0\widetilde{\Delta}^T\Sigma^{-1}\Delta_0}{1 + a_0\Delta^T\Sigma^{-1}\Delta_0}\Sigma^{-1}\Delta.$$

Let $c_0 = \frac{a_0 \widetilde{\Delta}^T \Sigma^{-1} \Delta_0}{1 + a_0 \Delta^T \Sigma^{-1} \Delta_0}$ and $c_1 = |c_0| \|\Delta\|_\Sigma / \|\widetilde{\Delta}\|_\Sigma$. For simplicity, let $\|v\|_\Sigma = v^T \Sigma^{-1} v$. We first lower bound it via

$$
\begin{aligned}
cor(b_{\mathrm{ERM}}, \widetilde{\Delta}) = \frac{b^T \widetilde{\Delta}}{\|\widetilde{\Delta}\|_\Sigma \sqrt{b^T \Sigma b}} &= \frac{\widetilde{\Delta}^T \Sigma^{-1} \widetilde{\Delta} - c_0 \Delta^T \Sigma^{-1} \widetilde{\Delta}}{\|\widetilde{\Delta}\|_\Sigma \sqrt{b^T \Sigma b}} \\
&\geq \frac{\widetilde{\Delta}^T \Sigma^{-1} \widetilde{\Delta}}{\|\widetilde{\Delta}\|_\Sigma (\|\widetilde{\Delta}\|_\Sigma + |c_0| \|\Delta\|_\Sigma)} - \frac{|c_0 \Delta^T \Sigma^{-1} \widetilde{\Delta}|}{\|\widetilde{\Delta}\|_\Sigma \sqrt{b^T \Sigma b}} \\
&\geq \frac{1}{1 + |c_0| \|\Delta\|_\Sigma / \|\widetilde{\Delta}\|_\Sigma} - \frac{c_0 \xi \|\Delta\|_\Sigma}{\|\widetilde{\Delta}\|_\Sigma - c_0 \|\Delta\|_\Sigma} \\
&\geq \frac{1 - (1 + \xi)c_1 - c_1^2}{1 - c_1^2} = 1 - C\alpha.
\end{aligned}
$$

Similarly, we have

$$
\begin{aligned}
cor(b_{\mathrm{ERM}}, \Delta) = \frac{b^T \Delta}{\|\Delta\|_\Sigma \sqrt{b^T \Sigma b}} &= \frac{\Delta^T \Sigma^{-1} \widetilde{\Delta} - c_0 \Delta^T \Sigma^{-1} \Delta}{\|\Delta\|_\Sigma \sqrt{b^T \Sigma b}} \\
&\leq \frac{\widetilde{\Delta}^T \Sigma^{-1} \Delta}{\|\Delta\|_\Sigma (\|\widetilde{\Delta}\|_\Sigma \pm c_0 \|\Delta\|_\Sigma)} + \frac{|c_0 \Delta^T \Sigma^{-1} \Delta|}{(\|\widetilde{\Delta}\|_\Sigma - c_0 \|\Delta\|_\Sigma) \|\Delta\|_\Sigma} \\
&\leq \frac{1}{1 \pm c_0 \|\Delta\|_\Sigma / \|\widetilde{\Delta}\|_\Sigma} \xi + \frac{c_0 \|\Delta\|_\Sigma / \|\widetilde{\Delta}\|_\Sigma}{1 - c_0 \|\Delta\|_\Sigma / \|\widetilde{\Delta}\|_\Sigma} \\
&\leq \left( \frac{\xi}{1 \pm c_1} - \frac{c_1}{1 - c_1} \right) \|\Delta\|_\Sigma.
\end{aligned}
$$

Hence,

$$
E_{\mathrm{ERM}}^{(wst)} \geq \max \left\{ \Phi\left( \left( \frac{1}{2} - C\alpha \right) \|\widetilde{\Delta}\|_\Sigma - (\xi - C\alpha) \|\Delta\|_\Sigma \right), \Phi\left( \left( -\frac{1}{2} + C\alpha \right) \|\widetilde{\Delta}\|_\Sigma \right) \right\} \tag{8}
$$

for some constant $C$ depending on the true parameters.

For method LISA-I, using the fact that $\Delta_0 = \Delta - \widetilde{\Delta}$,

$$
\begin{aligned}
b_{\mathrm{L1}} &\propto \left( 1 - \frac{a_{\mathrm{L1}} \Delta^T \Sigma^{-1} \Delta_0}{1 + a_{\mathrm{L1}} \Delta_0^T \Sigma^{-1} \Delta_0} \right) \Sigma^{-1} \Delta + \frac{a_{\mathrm{L1}} \Delta^T \Sigma^{-1} \Delta_0}{1 + a_{\mathrm{L1}} \Delta_0^T \Sigma^{-1} \Delta_0} \Sigma^{-1} \widetilde{\Delta} \\
&\propto \Sigma^{-1} \widetilde{\Delta} + c_{\mathrm{L1}} \Sigma^{-1} \Delta
\end{aligned}
$$

for

$$
c_{\mathrm{L1}} = \frac{1 + a_{\mathrm{L1}} \Delta_0^T \Sigma^{-1} \Delta_0 - a_{\mathrm{L1}} \Delta^T \Sigma^{-1} \Delta_0}{a_{\mathrm{L1}} \Delta^T \Sigma^{-1} \Delta_0}.
$$

Hence,

$$
cor(b_{\mathrm{L1}}, \widetilde{\Delta}) = \frac{\widetilde{\Delta}^T b_{\mathrm{L1}}}{\|\widetilde{\Delta}\|_\Sigma \sqrt{b_{\mathrm{L1}}^T \Sigma b_{\mathrm{L1}}}} = \frac{\|\widetilde{\Delta}\|_\Sigma + c_{\mathrm{L1}} \xi \|\Delta\|_\Sigma}{\|\widetilde{\Delta} + c_{\mathrm{L1}} \Delta\|_\Sigma}
$$

$$
cor(b_{\mathrm{L1}}, \Delta) = \frac{b_{\mathrm{L1}}^T \Delta}{\|\Delta\|_\Sigma \sqrt{b_{\mathrm{L1}}^T \Sigma b_{\mathrm{L1}}}} = \frac{\xi \|\widetilde{\Delta}\|_\Sigma \|\Delta\|_\Sigma + c_{\mathrm{L1}} \|\Delta\|_\Sigma^2}{\|\Delta\|_\Sigma \|\widetilde{\Delta} + c_{\mathrm{L1}} \Delta\|_\Sigma}.
$$

To have $E_{\mathrm{L1}}^{(wst)} \leq E_{\mathrm{ERM}}^{(wst)}$, it suffices to require that $\left( -\frac{1}{2} - C\alpha \right) \|\widetilde{\Delta}\|_\Sigma \leq \left( \frac{1}{2} - C\alpha \right) \|\widetilde{\Delta}\|_\Sigma - (\xi + C\alpha) \|\Delta\|_\Sigma$ and

$$
\frac{1}{2} cor(b_{\mathrm{L1}}, \widetilde{\Delta}) \|\widetilde{\Delta}\|_\Sigma - cor(b_{\mathrm{L1}}, \Delta) \|\Delta\|_\Sigma \leq \left( \frac{1}{2} - C\alpha \right) \|\widetilde{\Delta}\|_\Sigma - (\xi + C\alpha) \|\Delta\|_\Sigma
$$

$$
- \frac{1}{2} cor(b_{\mathrm{L1}}, \widetilde{\Delta}) \|\widetilde{\Delta}\|_\Sigma \leq \left( \frac{1}{2} - C\alpha \right) \|\widetilde{\Delta}\|_\Sigma - (\xi + C\alpha) \|\Delta\|_\Sigma.
$$

A sufficient condition is

$$\xi \le (\frac{1}{2} + \frac{1}{2}cor(b_{\mathrm{L1}}, \widetilde{\Delta}))\frac{\|\widetilde{\Delta}\|_\Sigma}{\|\Delta\|_\Sigma} - C\alpha, \ cor(b_{\mathrm{L1}}, \Delta) \ge \xi + C\alpha, \ cor(b_{\mathrm{L1}}, \widetilde{\Delta}) \le 1 - 2C\alpha.$$

We can find that a further sufficient condition is

$$\xi \le \frac{\|\widetilde{\Delta}\|_\Sigma}{\|\Delta\|_\Sigma} - C\alpha, c_{\mathrm{L1}} > 0, \xi \le \frac{\|\widetilde{\Delta} + c_{\mathrm{L1}}\Delta\|_\Sigma - \|\widetilde{\Delta}\|_\Sigma}{c_{\mathrm{L1}}\|\Delta\|_\Sigma} - \epsilon_1\alpha \tag{9}$$

$$\|\widetilde{\Delta} + c_{\mathrm{L1}}\Delta\|_\Sigma \ge \|\widetilde{\Delta}\|_\Sigma, \ \xi \le \frac{c_{\mathrm{L1}}\|\Delta\|_\Sigma}{\|\widetilde{\Delta} + c_{\mathrm{L1}}\Delta\|_\Sigma - \|\widetilde{\Delta}\|_\Sigma} - \epsilon_1\alpha \tag{10}$$

$$\xi \le (\frac{1}{2} + \frac{1}{2}cor(b_{\mathrm{L1}}, \widetilde{\Delta}))\frac{\|\widetilde{\Delta}\|_\Sigma}{\|\Delta\|_\Sigma} - C\alpha. \tag{11}$$

We first find sufficient conditions for the statements in (9) and (10). Parameterizing $t = c_{\mathrm{L1}}\|\Delta\|_\Sigma/\|\widetilde{\Delta}\|_\Sigma$, we further simplify the condition in (9) and (10) as

$$\xi \le \frac{\|\widetilde{\Delta}\|_\Sigma}{\|\Delta\|_\Sigma} - C\alpha, t > 0, \ -\frac{t}{2} \le \xi \le t$$

$$\xi \le \frac{\sqrt{1 + t^2 + 2t\xi} - 1}{t} - \epsilon_1\alpha, \ \xi \le \frac{1 + \sqrt{1 + t^2 + 2t\xi}}{t + 2\xi} - \epsilon_1\alpha.$$

We only need to require

$$t \ge 2 \text{ and } \xi \le \min\{\frac{\|\widetilde{\Delta}\|_\Sigma}{\|\Delta\|_\Sigma}, 1\} - C\alpha.$$

Some tedious calculation shows that $t \ge 2$ can be guaranteed by

$$\frac{1}{2}\|\Delta\|_\Sigma \le \|\widetilde{\Delta}\|_\Sigma \le \frac{1}{\sqrt{3a_{\mathrm{L1}}}} \text{ or } \frac{1}{2}\|\Delta\|_\Sigma \ge \|\widetilde{\Delta}\|_\Sigma.$$

It is left to consider the constraint in (11). Notice that it holds for any $\xi \le 0$. When $\xi > 0$, we can see

$$cor(b_{\mathrm{L1}}, \widetilde{\Delta}) = \frac{\|\widetilde{\Delta}\|_\Sigma + \xi c_{\mathrm{L1}}\|\Delta\|_\Sigma}{\|\widetilde{\Delta} + c_{\mathrm{L1}}\Delta\|_\Sigma} = \frac{1 + t\xi}{\sqrt{1 + t^2 + 2t\xi}}$$

$$\ge \frac{1 + t\xi}{1 + t} \ge \xi.$$

Hence, it suffices to guarantee that

$$(1 - \frac{1}{2}\frac{\|\widetilde{\Delta}\|_\Sigma}{\|\Delta\|_\Sigma})\xi < \frac{1}{2}\frac{\|\widetilde{\Delta}\|_\Sigma}{\|\Delta\|_\Sigma} - C\alpha.$$

If $\|\widetilde{\Delta}\|_\Sigma/\|\Delta\|_\Sigma \ge 2$, then LHS is negative and it holds. If $1 \le \|\widetilde{\Delta}\|_\Sigma/\|\Delta\|_\Sigma < 2$, then the inequality becomes $\xi \le 1 - C\alpha$. If $\|\widetilde{\Delta}\|_\Sigma/\|\Delta\|_\Sigma < 1$, then the inequality becomes $\xi \le \frac{\|\widetilde{\Delta}\|_\Sigma}{\|\Delta\|_\Sigma} - C\alpha$. Because we have required $\xi \le \min\{\frac{\|\widetilde{\Delta}\|_\Sigma}{\|\Delta\|_\Sigma}, 1\} - C\alpha$ for some large enough $C$, the constraint (11) always holds. To summarize, $E_{\mathrm{L1}} \le E_{\mathrm{ERM}}$ given that $\xi \le \min\{\frac{\|\widetilde{\Delta}\|_\Sigma}{\|\Delta\|_\Sigma}, 1\} - C\alpha$ for some large enough $C$ and $\|\widetilde{\Delta}\|_\Sigma \le \frac{1}{\sqrt{3a_{\mathrm{L1}}}}$.

For method LISA-II, we can similarly show that $E_{\mathrm{L2}} \le E_{\mathrm{ERM}}$ given that $\xi \le \min\{\frac{\|\widetilde{\Delta}\|_\Sigma}{\|\Delta\|_\Sigma}, 1\} - C\alpha$ for some large enough $C$ and $\|\widetilde{\Delta}\|_\Sigma \le \frac{1}{\sqrt{3a_{\mathrm{L2}}}}$.

## B.5 Finite sample analysis

The empirical loss can be written as

$$\mathbb{P}(\mathbb{1}((x_i^{G})^T\hat{b} + \hat{b}_0 > \frac{1}{2}) \ne y_i^{(G)}) \tag{12}$$

$$= \frac{1}{2}\mathbb{P}((x_i^{G})^T\hat{b} + \hat{b}_0 > \frac{1}{2}|y_i^{(G)} = 0) + \frac{1}{2}\mathbb{P}((x_i^{G})^T\hat{b} + \hat{b}_0 < \frac{1}{2}|y_i^{(G)} = 1),$$

where

$$\mathbb{P}((x_i^{G})^T\hat{b} + \hat{b}_0 > \frac{1}{2}|y_i^{(G)} = 0) = \Phi(-\frac{\frac{1}{2} - (\mu^{(0,G)})^T\hat{b} - \hat{b}_0}{\sqrt{\hat{b}^T\Sigma\hat{b}}}).$$

$$\mathbb{P}((x_i^{G})^T\hat{b} + \hat{b}_0 < \frac{1}{2}|y_i^{(G)} = 1) = \Phi(\frac{\frac{1}{2} - (\mu^{(1,G)})^T\hat{b} - \hat{b}_0}{\sqrt{\hat{b}^T\Sigma\hat{b}}}).$$

First notice that

$$\hat{b}_0 = \bar{y} - \bar{x}^T\hat{b}.$$

We have

$$(\mu^{(0,G)})^T\hat{b} + \hat{b}_0 = (\mu^{(0,G)} - \bar{x})^T\hat{b} + \bar{y}$$
$$= (\mu^{(0,G)} - \mathbb{E}[x_i])^T\hat{b} + \frac{1}{2} + \underbrace{\{(\bar{y} - \bar{x}^T\hat{b}) - (\mathbb{E}[y_i] - \mathbb{E}[x_i]^T\hat{b})\}}_{R_1}$$

$$(\mu^{(1,G)})^T\hat{b} + \hat{b}_0 = (\mu^{(1,G)} - \bar{x})^T\hat{b} + \bar{y}$$
$$= \Delta^T\hat{b} + (\mu^{(0,G)} - \mathbb{E}[x_i])^T\hat{b} + \frac{1}{2} + R_1.$$

Therefore, according to (12),

$$\frac{1}{2}\Phi(-\frac{\frac{1}{2} - (\mu^{(0,G)})^T\hat{b} - \hat{b}_0}{\sqrt{\hat{b}^T\Sigma\hat{b}}}) + \frac{1}{2}\Phi(\frac{\frac{1}{2} - (\mu^{(1,G)})^T\hat{b} - \hat{b}_0}{\sqrt{\hat{b}^T\Sigma\hat{b}}})$$

$$=\frac{1}{2}\Phi(\frac{(\theta^{(G)})^T\hat{b} + R_1}{\sqrt{\hat{b}^T\Sigma\hat{b}}}) + \frac{1}{2}\Phi(-\frac{\Delta + (\theta^{(G)})^T\hat{b} + R_1}{\sqrt{\hat{b}^T\Sigma\hat{b}}})$$

$$=\frac{1}{2}\Phi(\frac{(\theta^{(G)})^T\hat{b} + R_1}{\sqrt{\hat{b}^T\Sigma\hat{b}}}) + \frac{1}{2}\Phi(-\frac{(\theta^{(G)})^T\hat{b} + R_1}{\sqrt{\hat{b}^T\Sigma\hat{b}}})$$

$$-\left\{\frac{1}{2}\Phi(-\frac{(\theta^{(G)})^T\hat{b} + R_1}{\sqrt{\hat{b}^T\Sigma\hat{b}}}) - \frac{1}{2}\Phi(-\frac{\Delta + (\Theta^{(G)})^T\hat{b} + R_1}{\sqrt{\hat{b}^T\Sigma\hat{b}}})\right\}$$

$$=\frac{1}{2} - \left\{\frac{1}{2}\Phi(-\frac{(\theta^{(G)})^T\hat{b} + R_1}{\sqrt{\hat{b}^T\Sigma\hat{b}}}) - \frac{1}{2}\Phi(-\frac{\Delta^T\hat{b} + (\theta^{(G)})^T\hat{b} + R_1}{\sqrt{\hat{b}^T\Sigma\hat{b}}})\right\}.$$

Then the mis-classification error can be written as

$$\frac{1}{2} - \frac{1}{2}\underbrace{\left\{\Phi(\frac{(\theta^{(G)})^T\hat{b} + R_1}{\sqrt{\hat{b}^T\Sigma\hat{b}}}) - \Phi(\frac{(\theta^{(G)})^T\hat{b} - \Delta^T\hat{b} + R_1}{\sqrt{\hat{b}^T\Sigma\hat{b}}})\right\}}_{\widehat{L}(\hat{b})}. \tag{13}$$

Larger the $\widehat{L}(\hat{b})$, smaller the mis-classification error.

We first find that

$$\widehat{L}(\hat{b}) - L(b) \le C \underbrace{|\frac{(\theta^{(G)})^T\hat{b} + R_1}{\sqrt{\hat{b}^T\Sigma\hat{b}}} - \frac{(\theta^{(G)})^Tb}{\sqrt{b^T\Sigma b}}|}_{T_1} + C \underbrace{|\frac{(\theta^{(G)})^T\hat{b} - \Delta^T\hat{b} + R_1}{\sqrt{\hat{b}^T\Sigma\hat{b}}} - \frac{(\theta^{(G)})^Tb - \Delta^Tb}{\sqrt{b^T\Sigma b}}|}_{T_2}.$$

In the event that

$$\|\Sigma^{1/2}(\hat{b} - b)\|_2 = o(1) \max_{y,d}\|\mu^{(y,d)}\|_2 \le C, \ \Sigma \text{ is positive definite.}$$

for the denominator, we have

$$|b^T \Sigma b - \hat{b}^T \Sigma \hat{b}| \leq (2\|\Sigma^{1/2}b\|_2 + \|\Sigma^{1/2}(\hat{b} - b)\|_2)\|\Sigma^{1/2}(\hat{b} - b)\|_2$$

$$\leq 2(1 + o(1))\|\Sigma^{1/2}b\|_2\|\Sigma^{1/2}(\hat{b} - b)\|_2$$

$$|\sqrt{\hat{b}^T \Sigma \hat{b}} - \sqrt{b^T \Sigma b}| \leq \frac{|\hat{b}^T \Sigma \hat{b} - b^T \Sigma b|}{\sqrt{\hat{b}^T \Sigma \hat{b}} + \sqrt{b^T \Sigma b}}$$

$$\leq 2(1 + o(1))\|\Sigma^{1/2}(\hat{b} - b)\|_2.$$

For the numerator, we have

$$|\frac{1}{2}\widetilde{\Delta}^T \hat{b} + R_1 - \frac{1}{2}\widetilde{\Delta}^T b| \leq |R_1| + \frac{1}{2}\|\Sigma^{-1/2}\widetilde{\Delta}\|_2\|\Sigma^{1/2}(\hat{b} - b)\|_2.$$

We arrive at

$$T_1 \leq (1 + o(1))\frac{|R_1| + \frac{1}{2}\|\Sigma^{-1/2}\widetilde{\Delta}\|_2\|\Sigma^{1/2}(\hat{b} - b)\|_2}{\|\Sigma^{1/2}b\|_2} + (1 + o(1))\frac{|\widetilde{\Delta}^T b|}{\sqrt{b^T \Sigma b}}\frac{\|\Sigma^{1/2}(\hat{b} - b)\|_2}{\sqrt{b^T \Sigma b}}.$$

$$T_2 \leq (1 + o(1))\frac{|R_1| + \frac{1}{2}(\|\Sigma^{-1/2}\widetilde{\Delta}\|_2 + \|\Sigma^{-1/2}\Delta\|_2)\|\Sigma^{1/2}(\hat{b} - b)\|_2}{\|\Sigma^{1/2}b\|_2}$$

$$+ (1 + o(1))\frac{|\frac{1}{2}\widetilde{\Delta}^T b - \Delta^T b|}{\sqrt{b^T \Sigma b}}\frac{\|\hat{b} - b\|_2}{\sqrt{b^T \Sigma b}}.$$

Moreover $R_1 \leq \|\hat{b} - b\|_2 + O_P(\frac{1}{\sqrt{n}})$. To summarize,

$$|\widehat{L}(\hat{b}) - L(b)| \lesssim (1 + o(1))(\|\hat{b} - b\|_2 + \frac{1}{\sqrt{n}}).$$

In the following, we will upper bound $\|\hat{b} - b\|_2$ for each method. For the **ERM method**,

$$\hat{b} = \{(X - \bar{X})^T(X - \bar{X})\}^{-1}(X - \bar{X})^T(y - \bar{y}).$$

It is easy to show that

$$\|\hat{b} - b\|_2^2 = O_P(\frac{p \sum_{i=1}^N var(y_i|x_i)}{N^2}) = O_P(\frac{p}{N}).$$

For the **vanilla mixup method**, we first see that

$$\frac{1}{n}\sum_{i=1}^n \tilde{x}_i = \frac{1}{n}\sum_{i=1}^n (\lambda_i x_{i_1} + (1 - \lambda_i)x_{i_2}) = \bar{x} + O_P(n^{-1/2}) = \mu + O_P(n^{-1/2})$$

$$\frac{1}{n}\sum_{i=1}^n \tilde{y}_i = \pi^{(1)} + O_P(n^{-1/2}).$$

Next,

$$\frac{1}{n}\sum_{i=1}^n \tilde{x}_i\tilde{y}_i = \frac{1}{n}\sum_{i=1}^n \{\lambda_i^2 x_{i_1}y_{i_1} + (1 - \lambda_i)^2 x_{i_2}y_{i_2} + \lambda_i(1 - \lambda_i)x_{i_1}y_{i_2} + \lambda_i(1 - \lambda_i)x_{i_2}y_{i_i}\}$$

$$\frac{1}{n}\sum_{i=1}^n \tilde{x}_i\tilde{y}_i - \mathbb{E}[\tilde{x}_i\tilde{y}_i] = \underbrace{\frac{1}{n}\sum_{i=1}^n \tilde{x}_i\tilde{y}_i - \mathbb{E}[\tilde{x}_i\tilde{y}_i|X, y]}_{E_1} + \underbrace{\mathbb{E}[\tilde{x}_i\tilde{y}_i|X, y] - \mathbb{E}[\tilde{x}_i\tilde{y}_i]}_{E_2}.$$

For $E_2$,

$$E_2 = \frac{2\mathbb{E}[\lambda_i^2]}{n}\sum_{i=1}^n x_iy_i - \mathbb{E}[\tilde{x}_i\tilde{y}_i] = 2\mathbb{E}[\lambda_i^2]\mathbb{E}[x_iy_i].$$

Hence,

$$\|E_2\|_2^2 = O_P(\frac{p}{n}).$$

For $E_1$, conditioning on $(X, y)$, $\lambda_i^2 x_{i_1} y_{i_1} - \frac{\mathbb{E}[\lambda_i^2]}{n} \sum_{i=1}^n x_i y_i$ are independent sub-Gaussian vectors. The sub-Gaussian norm of $\frac{1}{N} \sum_{i=1}^n \lambda_i^2 x_{i_1,j} y_{i_1} - \frac{\mathbb{E}[\lambda_i^2]}{n} \sum_{i=1}^n x_{i,j} y_i$ (conditioning on $(X, y)$) can be upper bounded by $c \max_{i \leq N} |x_{i,j}| / \sqrt{n}$. Hence

$$\mathbb{P}(\|E_1\|_2 \geq t | X, y) \leq 2 \exp\{-\frac{c_2 n t^2}{\max_{j=1}^p \max_{i \leq N} x_{i,j}^2}\}.$$

As $x_{i,j}$ are Gaussian distributed, we know that

$$\mathbb{P}(\sum_{j=1}^p \max_{i \leq n} x_{i,j}^2 \geq p \log n) \leq \exp\{-c_3 \log n\}.$$

Hence, with probability at least $1 - \exp(-c_1 \log n)$,

$$E_1 \leq \frac{Cp \log n}{n}.$$

To summarize,

$$\left\| \frac{1}{n} \sum_{i=1}^n \tilde{x}_i \tilde{y}_i - (\frac{1}{n} \sum_{i=1}^n \tilde{x}_i)(\frac{1}{n} \sum_{i=1}^n \tilde{y}_i) - cov(\tilde{x}_i, \tilde{y}_i) \right\|_2^2 = O_P(\frac{p \log n}{n}).$$

Similarly, we can show that

$$\left\| \frac{1}{n} \sum_{i=1}^n \tilde{x}_i \tilde{x}_i^T - (\frac{1}{n} \sum_{i=1}^n \tilde{x}_i)(\frac{1}{n} \sum_{i=1}^n \tilde{x}_i)^T - cov(\tilde{x}_i) \right\|_2^2 = O_P(\frac{p \log n}{n}).$$

Hence,

$$\|\hat{b} - b\|_2^2 = O_P(\frac{p \log n}{n}).$$

For the **LISA-I**, we first see that

$$\frac{1}{n} \sum_{i=1}^n x_i^{(\lambda)} = \frac{1}{n} \sum_{y_i=1} (\lambda_i x_{i_1}^{(1,G)} + (1 - \lambda_i) x_{i_2}^{(1,B)}) + \frac{1}{n} \sum_{y_i=0} (\lambda_i x_{i_1}^{(0,G)} + (1 - \lambda_i) x_{i_2}^{(0,B)})$$

$$= \frac{1}{2} (\bar{x}^{(1,G)} + \bar{x}^{(1,B)}) \hat{\pi}_1 + \frac{1}{2} (\bar{x}^{(0,G)} + \bar{x}^{(0,B)}) \hat{\pi}_0$$

We have

$$\frac{1}{n}(X^{(\lambda)})^T y - \bar{y} \frac{1}{n} \sum_{i=1}^n x_i^{(\lambda)} - cov(x_i^{(\lambda)}, y_i) = \underbrace{\frac{1}{n}(X^{(\lambda)})^T y - \bar{y} \frac{1}{n} \sum_{i=1}^n x_i^{(\lambda)} - cov(x_i^{(\lambda)}, y_i | X, y)}_{E_1}$$

$$+ \underbrace{cov(x_i^{(\lambda)}, y_i | X, y) - cov(x_i^{(\lambda)}, y_i)}_{E_2}$$

For $E_2$,

$$E_2 = \frac{\hat{\pi}_1}{2} (\bar{x}^{(1,G)} + \bar{x}^{(1,B)}) - \hat{\pi}_1 (\frac{1}{2} (\bar{x}^{(1,G)} + \bar{x}^{(1,B)}) \hat{\pi}_1 + \frac{1}{2} (\bar{x}^{(0,G)} + \bar{x}^{(0,B)}) \hat{\pi}_0) - cov(x_i^{(\lambda)}, y_i)$$

$$= \frac{1}{2} (\bar{x}^{(1,G)} + \bar{x}^{(1,B)} - \bar{x}^{(0,G)} - \bar{x}^{(0,B)}) \hat{\pi}_1 \hat{\pi}_0 - \pi^{(1)} \pi^{(0)} \Delta.$$

It is easy to show that

$$\|E_2\|_2^2 = O_P \left( \frac{p}{\min_{y,e} n^{(y,e)}} \right).$$

For $E_1$, conditioning on $X$ and $y$, $x_i^{(\lambda)} y_i - \mathbb{E}[x_i^{(\lambda)} y_i | X, y]$ are independent sub-Gaussian vectors with mean zero. The sub-Gaussian norm of $\frac{1}{n} \sum_{i=1}^n x_{i,j}^{(\lambda)} y_i$ (conditioning on $X$ and $y$) can be upper bounded by $c \max_{i \le n} |x_{i,j}| / \sqrt{N}$.

$$\mathbb{P}(\|E_1\|_2 \ge t | X, y) = \mathbb{P}\left( \sum_{j=1}^p | \frac{1}{n} \sum_{i=1}^n \{x_{i,j}^{(\lambda)} y_i - \mathbb{E}[x_{i,j}^{(\lambda)} y_i | X, y]\}|^2 \ge t^2 | X, y \right)$$

$$\le 2 \exp\left\{ -\frac{c_2 n t^2}{\sum_{j=1}^p \max_{i \le n} x_{i,j}^2} \right\}.$$

Hence,

$$E_1 = O_P(\sqrt{\frac{\sum_{j=1}^p \max_{i \le n} x_{i,j}^2}{n}}) = O_P(\frac{p \log n}{n}).$$

To summarize,

$$\|\frac{1}{n}(X^{(\lambda)})^T y - \mathbb{E}[x_i^{(\lambda)} y_i]\|_2^2 = O_P(\frac{p}{\min_{y,e} n^{(y,e)}} + \frac{p \log n}{n}).$$

We can use similar analysis to bound

$$\|\frac{1}{N}(X^{(\lambda)})^T X^{(\lambda)} - \mathbb{E}[x_i^{(\lambda)} (x_i^{(\lambda)})^T]\|_2.$$

The sub-exponential norm of $\frac{1}{N} \sum_{i=1}^N x_{i,j}^{(\lambda)} x_{i,k}^{(\lambda)}$ (conditioning on $X$) can be upper bounded by $\max_{i \le N} |x_{i,j}||x_{i,k}| / \sqrt{N}$. We can show that

$$\|\frac{1}{n}(X^{(\lambda)})^T X^{(\lambda)} - \mathbb{E}[x_i^{(\lambda)} (x_i^{(\lambda)})^T]\|_2 = O_P(\frac{p}{\min_{y,e} n^{(y,e)}} + \frac{p \log n}{n}).$$

For the **LISA-II**, we first see that

$$\frac{1}{n} \sum_{i=1}^n \tilde{x}_i = \frac{1}{n} \sum_{D_i=G} (\lambda_i x_{i_1}^{(1,G)} + (1-\lambda_i) x_{i_2}^{(0,G)}) + \frac{1}{n} \sum_{D_i=B} (\lambda_i x_{i_1}^{(1,B)} + (1-\lambda_i) x_{i_2}^{(0,B)})$$

$$= \frac{1}{2}(\bar{x}^{(1,G)} + \bar{x}^{(0,G)})\hat{\pi}_G + \frac{1}{2}(\bar{x}^{(1,B)} + \bar{x}^{(0,B)})\hat{\pi}_B$$

$$\bar{\tilde{y}} = \frac{1}{2}.$$

Next,

$$\frac{1}{n} \sum_{i=1}^n \tilde{x}_i \tilde{y}_i = \frac{1}{n} \sum_{D_i=G} \left\{ \lambda_i^2 x_{i_1}^{(1,G)} + \lambda_i(1-\lambda_i) x_{i_2}^{(0,G)} \right\} + \frac{1}{n} \sum_{D_i=B} \left\{ \lambda_i^2 x_{i_1}^{(1,B)} + \lambda_i(1-\lambda_i) x_{i_2}^{(0,B)} \right\}$$

$$\frac{1}{n} \sum_{i=1}^n \tilde{x}_i \tilde{y}_i - \bar{\tilde{x}}\bar{\tilde{y}} - cov(\tilde{x}, \tilde{y}) = \underbrace{\frac{1}{n} \sum_{i=1}^n \tilde{x}_i \tilde{y}_i - \bar{\tilde{x}}\bar{\tilde{y}} - cov(\tilde{x}_i, \tilde{y}_i | X, y)}_{E_1} + \underbrace{cov(\tilde{x}_i, \tilde{y}_i | X, y) - cov(\tilde{x}_i, \tilde{y}_i)}_{E_2}.$$

For $E_2$,

$$E_2 = \hat{\pi}^{(G)}(\mathbb{E}[\lambda_i^2](\bar{x}^{(1,G)} - \bar{x}^{(0,G)}) + \frac{1}{2}\bar{x}^{(0,G)}) + \hat{\pi}^{(B)}(\mathbb{E}[\lambda_i^2](\bar{x}^{(1,B)} - \bar{x}^{(0,B)}) + \frac{1}{2}\bar{x}^{(0,B)}) -$$

$$\frac{1}{4}(\bar{x}^{(1,G)} + \bar{x}^{(0,G)})\hat{\pi}_G - \frac{1}{4}(\bar{x}^{(1,B)} + \bar{x}^{(0,B)})\hat{\pi}_B - var(\lambda_i)\Delta$$

$$= \hat{\pi}^{(G)} var(\lambda_i)(\bar{x}^{(1,G)} - \bar{x}^{(0,G)}) + \hat{\pi}^{(B)} var(\lambda_i)(\bar{x}^{(1,B)} - \bar{x}^{(0,B)}) - var(\lambda_i)\Delta.$$

Notice that $E_2$ is a sub-Gaussian vector with sub-Gaussian norm upper bounded by

$$\frac{\hat{\pi}_G^2}{n^{(1,G)}} + \frac{\hat{\pi}_G^2}{n^{(0,G)}} + \frac{\hat{\pi}_B^2}{n^{(1,B)}} + \frac{\hat{\pi}_B^2}{n^{(0,B)}} \le \frac{4}{n} \max_{y,d} \frac{\pi_d}{\pi_{y|d}}.$$

Using sub-Gaussian concentration, we can show that

$$E_2 = O_P\left(\sqrt{\frac{p}{n} \max_{y,d} \frac{\pi_d}{\pi_{y|d}}}\right).$$

Notice that $\max_{y,d} \frac{\pi_d}{\pi_{y|d}} \geq 1$. For $E_1$, conditioning on $X$ and $y$ $\tilde{x}_i\tilde{y}_i - \mathbb{E}[\tilde{x}_i\tilde{y}_i|X,y]$ are independent sub-Gaussian vectors with mean zero. The sub-Gaussian norm of $\frac{1}{n}\sum_{i=1}^n \tilde{x}_{i,j}\tilde{y}_i$ conditioning on $X$ and $y$ can be upper bounded by $c\max_{i,j}|x_{i,j}|$. Similar analysis on $E_1$ leads to

$$\frac{1}{n}\sum_{i=1}^n \tilde{x}_i\tilde{y}_i - \bar{\tilde{x}}\bar{\tilde{y}} - cov(\tilde{x},\tilde{y}) = O_P\left(\sqrt{\frac{p\log n}{n}} + \sqrt{\frac{p}{n}\max_{y,d}\frac{\pi_d}{\pi_{y|d}}}\right).$$

For the sample covariance matrix, we can also show that

$$\left\|\frac{1}{n}\sum_{i=1}^n \tilde{x}_i\tilde{x}_i^T - \left(\frac{1}{n}\sum_{i=1}^n \tilde{x}_i\right)\left(\frac{1}{n}\sum_{i=1}^n \tilde{x}_i\right)^T - cov(\tilde{x}_i)\right\|_2^2 = O_P\left(\sqrt{\frac{p\log n}{n}} + \sqrt{\frac{p}{n}\max_{y,d}\frac{\pi_d}{\pi_{y|d}}}\right).$$

## B.6 DOMAIN SHIFTS: PROOF OF THEOREM 2

It still holds that $\widetilde{\Delta}^* = 2(\mu^{(0,*)} - \mathbb{E}[x_i^{(\lambda)}]) = 2(\mu^{(0,*)} - \mathbb{E}[\tilde{x}_i])$. It is easy to show that the worst group mis-classification error for this new environment is

$$E_A^{(wst,*)} = \max\left\{\Phi\left(\frac{\frac{1}{2}(\widetilde{\Delta}^*)^T b_A}{\sqrt{b_A^T \Sigma b_A}}\right), \Phi\left(\frac{\frac{1}{2}(\widetilde{\Delta}^*)^T b_A - \Delta^T b_A}{\sqrt{b_A^T \Sigma b_A}}\right)\right\}, \tag{14}$$

where $A \in \{\text{ERM}, \text{mix}, \text{L1}, \text{L2}\}$. Notice that

$$\widetilde{\Delta}^* = 2\mu^{(0,*)} - (\mu^{(0,G)} + \mu^{(1,B)}) = \widetilde{\Delta} + \mu^{(0,*)} - \mu^{(0,G)}$$

We assume $\|\widetilde{\Delta}^*\|_2 = \|\widetilde{\Delta}\|_2$. Let $\xi^* = cor(\Delta, \widetilde{\Delta}^*)$ and $\gamma = cor(\widetilde{\Delta}, \widetilde{\Delta}^*)$. We have

$$cor(b_{\text{ERM}}, \widetilde{\Delta}^*) = \frac{\gamma\|\widetilde{\Delta}\|_\Sigma\|\widetilde{\Delta}^*\|_\Sigma - c_0\xi^*\|\Delta\|_\Sigma\|\widetilde{\Delta}^*\|_\Sigma}{\|\widetilde{\Delta}^*\|_\Sigma\|\widetilde{\Delta} + c_0\Delta\|_\Sigma}$$

$$= \frac{\gamma\|\widetilde{\Delta}\|_\Sigma}{\|\widetilde{\Delta}\|_\Sigma \pm \|c_0\Delta\|_\Sigma} \pm \frac{|c_0\xi^*|\|\Delta\|_\Sigma}{\|\widetilde{\Delta}\|_\Sigma \pm \|c_0\Delta\|_\Sigma} = \gamma \pm C\alpha.$$

Hence,

$$E_{\text{ERM}}^{(wst)} \geq \max\left\{\Phi\left((\frac{\gamma}{2} - C\alpha)\|\widetilde{\Delta}\|_\Sigma - (\xi - C\alpha)\|\Delta\|_\Sigma\right), \Phi\left((-\frac{\gamma}{2} - C\alpha)\|\widetilde{\Delta}\|_\Sigma\right)\right\} \tag{15}$$

for some constant $C$ depending on the true parameters.

Hence,

$$cor(b_{\text{L1}}, \widetilde{\Delta}^*) = \frac{(\widetilde{\Delta}^*)^T b_{\text{L1}}}{\|\widetilde{\Delta}^*\|_\Sigma\sqrt{b_{\text{L1}}^T \Sigma b_{\text{L1}}}} = \frac{\gamma\|\widetilde{\Delta}\|_\Sigma + c_{\text{L1}}\xi^*\|\Delta\|_\Sigma}{\|\widetilde{\Delta} + c_{\text{L1}}\Delta\|_\Sigma}.$$

To have $E_{\text{L1}}^{(wst*)} \leq E_{\text{ERM}}^{(wst*)}$, it suffices to require that $(-\frac{\gamma}{2} - C\alpha)\|\widetilde{\Delta}\|_\Sigma \leq (\frac{\gamma}{2} - C\alpha)\|\widetilde{\Delta}\|_\Sigma - (\xi + C\alpha)\|\Delta\|_\Sigma$ and

$$\frac{1}{2}cor(b_{\text{L1}}, \widetilde{\Delta}^*)\|\widetilde{\Delta}\|_\Sigma - cor(b_{\text{L1}}, \Delta)\|\Delta\|_\Sigma \leq (\frac{\gamma}{2} - C\alpha)\|\widetilde{\Delta}\|_\Sigma - (\xi + C\alpha)\|\Delta\|_\Sigma$$

$$-\frac{1}{2}cor(b_{\text{L1}}, \widetilde{\Delta}^*)\|\widetilde{\Delta}\|_\Sigma \leq (\frac{\gamma}{2} - C\alpha)\|\widetilde{\Delta}\|_\Sigma - (\xi + C\alpha)\|\Delta\|_\Sigma.$$

A sufficient condition is

$$\xi \leq (\frac{\gamma}{2} + \frac{1}{2}cor(b_{\text{L1}}, \widetilde{\Delta}^*))\frac{\|\widetilde{\Delta}\|_\Sigma}{\|\Delta\|_\Sigma} - C\alpha, \quad cor(b_{\text{L1}}, \Delta) \geq \xi + C\alpha, \quad cor(b_{\text{L1}}, \widetilde{\Delta}^*) \leq \gamma - 2C\alpha.$$

We can find that a further sufficient condition is

$$\xi \le \frac{1+\gamma}{2}\frac{\|\widetilde{\Delta}\|_\Sigma}{\|\Delta\|_\Sigma} - C\alpha, c_{\text{L1}} > 0, \xi^* \le \frac{\gamma(\|\widetilde{\Delta} + c_{\text{L1}}\Delta\|_\Sigma - \|\widetilde{\Delta}\|_\Sigma)}{c_{\text{L1}}\|\Delta\|_\Sigma} - \epsilon_1\alpha \tag{16}$$

$$\|\widetilde{\Delta} + c_{\text{L1}}\Delta\|_\Sigma \ge \|\widetilde{\Delta}\|_\Sigma, \xi \le \frac{c_{\text{L1}}\|\Delta\|_\Sigma}{\|\widetilde{\Delta} + c_{\text{L1}}\Delta\|_\Sigma - \|\widetilde{\Delta}\|_\Sigma} - \epsilon_1\alpha \tag{17}$$

$$\xi \le (\frac{\gamma}{2} + \frac{1}{2}cor(b_{\text{L1}}, \widetilde{\Delta}^*))\frac{\|\widetilde{\Delta}\|_\Sigma}{\|\Delta\|_\Sigma} - C\alpha. \tag{18}$$

We first find sufficient conditions for the statements in (9) and (10). Parameterizing $t = c_{\text{L1}}\|\Delta\|_\Sigma/\|\widetilde{\Delta}\|_\Sigma$, we further simplify the condition in (16) and (17) as

$$\xi \le \frac{1+\gamma}{2}\frac{\|\widetilde{\Delta}\|_\Sigma}{\|\Delta\|_\Sigma} - C\alpha, t > 0,\ \xi^* \le \frac{\gamma(\sqrt{1 + t^2 + 2t\xi} - 1)}{t} - \epsilon_1\alpha,$$

$$-\frac{t}{2} \le \xi \le t,\ \ \xi \le \frac{1 + \sqrt{1 + t^2 + 2t\xi}}{t + 2\xi} - \epsilon_1\alpha.$$

We only need to require

$$t \ge 2 \text{ and } \xi \le \min\{\frac{1+\gamma}{2}\frac{\|\widetilde{\Delta}\|_\Sigma}{\|\Delta\|_\Sigma}, 1\} - C\alpha, \xi^* \le \gamma\xi.$$

Some tedious calculation shows that $t \ge 2$ can be guaranteed by

$$\frac{1}{2}\|\Delta\|_\Sigma \le \|\widetilde{\Delta}\|_\Sigma \le \frac{1}{\sqrt{3a_{\text{L1}}}} \text{ or } \frac{1}{2}\|\Delta\|_\Sigma \ge \|\widetilde{\Delta}\|_\Sigma.$$

It is left to consider the constraint in (18). Notice that it holds for any $\xi \le 0$. When $\xi > 0$, we can see

$$cor(b_{\text{L1}}, \widetilde{\Delta}^*) = \frac{\gamma\|\widetilde{\Delta}\|_\Sigma + \xi^* c_{\text{L1}}\|\Delta\|_\Sigma}{\|\widetilde{\Delta} + c_{\text{L1}}\Delta\|_\Sigma} = \frac{\gamma + t\xi^*}{\sqrt{1 + t^2 + 2t\xi}}$$
$$\ge \frac{\gamma + t\xi^*}{1 + t}.$$

Hence, it suffices to guarantee that

$$\xi^* + \gamma \ge \frac{2\|\Delta\|_\Sigma}{\|\widetilde{\Delta}\|_\Sigma}\xi + C\alpha.$$

To summarize, it suffices to require

$$\|\widetilde{\Delta}\|_\Sigma \le \frac{1}{\sqrt{3a_{\text{L1}}}}, 0 \le \xi^* \le \gamma\xi, \xi \le \min\{\frac{\gamma}{2}\frac{\|\widetilde{\Delta}\|_\Sigma}{\|\Delta\|_\Sigma}, 1\} - C\alpha.$$

For LISA-II, we can similarly show that $E_{\text{L2}}^{(wst*)} \le E_{\text{ERM}}^{(wst*)}$ given that

$$\|\widetilde{\Delta}\|_\Sigma \le \frac{1}{\sqrt{3a_{\text{L2}}}}, 0 \le \xi^* \le \gamma\xi, \xi \le \min\{\frac{\gamma}{2}\frac{\|\widetilde{\Delta}\|_\Sigma}{\|\Delta\|_\Sigma}, 1\} - C\alpha.$$

