# OpenReview forum: "Improving Out-of-Distribution Robustness via Selective Augmentation"
_ICLR.cc/2022/Conference — ICLR 2022 Submitted_

### Official Review · Reviewer_4YgJ · 2021-10-18

**Correctness:** 2
**Technical Novelty And Significance:** 3
**Empirical Novelty And Significance:** 3
**Recommendation:** 5
**Confidence:** 3

**Main Review:**

Strengths:

+ The proposed approach is simple and efficient, and can be directly incorporated in or combined with other invariance-inducing approaches;

+ Prediction performance is shown to improve over a number of recent baselines under challenging benchmarks.

Weaknesses/suggestions:

- The proposal requires assumptions that are not discussed in the manuscript. In [1], it was shown that domain-invariant approaches can only improve out-of-distribution generalization if data-conditional label distributions P(y|x) are fixed across domains; i.e., observing x suffices in order to determine y, regardless of the domain according to which x was observed.

- My main concern lies in the reported evaluation, which is focused on showing improvements in terms of downstream performance, and presented results consist of comparing the proposed approach with alternative methods. While improving downstream performance is of course our ultimate goal, and results are strong in this sense, doing so does not explain the sources of improvements. In particular, authors claim that the mixup strategies they introduce yield some type of domain invariance, which is not verified empirically. To verify that learned representations are invariant, that could be achieved via domain-prediction experiments; i.e., train domain classifiers on top of representations learned by different methods. The higher the accuracy of such a classifier, the less invariant are representations. For the case of prediction-level invariance, authors could perhaps evaluate the range of estimated risks across domains. Improvements on either one of these notions of invariance would then explain observed improvements in terms of prediction accuracy.

- "We argue that the failure of other methods in some datasets may be caused by their regularizers limiting model capacity to some extent". That's another case where the evaluation lacks in supporting authors' claims. This hypothesis can be verified via in-domain prediction performance, i.e., overly regularized underfitting models should result in accuracy degradation in the training domains. Alternatively, one could rule out the underfitting effect by using higher capacity model classes.

- Conclusions from the risk bounds provided in theorems 1 and 2 are a bit unrealistic given the strong assumptions that imply the results. In particular, the model assumed for the data generating process is overly simplified and, although it enables theoretical analysis, it's unclear to which extend the conclusions hold in practice.

- Finally, regarding novelty, it seems different recent approaches introduce methods that use some sort of mixup across domains in similar settings. For the domain adaptation/generalization cases, there are, for instance, [2,3,4]. For the multi-domain case, cross-domain mixup was studied in [5]. Authors do compare results against [4], but it's unclear how the proposed approach differs from those other recent applications of mixup under similar settings, and the related work section should include such a discussion.

Other comments:

- On page 2, the setting described was not originally introduced by Koh et al. (2021). To my knowledge, it was first discussed in [6] and later on in [7].

- The definition of mixup for labels in the rightmost term in eq. 2 seems to require labels y are one-hot encoded, which is not mentioned in the text.

- While in the title authors claim to be improving out-of-distribution robustness, a large part of the work focus on the multi-domain learning (or fairness) setting, where one's goal is to find predictors with uniform risks across a set of domains that doesn't change from training to testing. Technically, that wouldn't be out-of-distribution. Perhaps there should be a sentence or two in the introduction indicating what authors refer to as out-of-distribution.

References:

[1] Zhao, Han, et al. "On learning invariant representations for domain adaptation." International Conference on Machine Learning. PMLR, 2019.

[2] Shu, Yang, et al. "Open Domain Generalization with Domain-Augmented Meta-Learning." Proceedings of the IEEE/CVF Conference on Computer Vision and Pattern Recognition. 2021.

[3] Wang, Yufei, Haoliang Li, and Alex C. Kot. "Heterogeneous domain generalization via domain mixup." ICASSP 2020-2020 IEEE International Conference on Acoustics, Speech and Signal Processing (ICASSP). IEEE, 2020.

[4] Xu, Minghao, et al. "Adversarial domain adaptation with domain mixup." Proceedings of the AAAI Conference on Artificial Intelligence. Vol. 34. No. 04. 2020.

[5] Chuang, Ching-Yao, and Youssef Mroueh. "Fair mixup: Fairness via interpolation." arXiv preprint arXiv:2103.06503 (2021).

[6] Muandet, Krikamol, David Balduzzi, and Bernhard Schölkopf. "Domain generalization via invariant feature representation." International Conference on Machine Learning. PMLR, 2013.

[7] Albuquerque, Isabela, et al. "Generalizing to unseen domains via distribution matching." arXiv preprint arXiv:1911.00804 (2019).


**Summary Of The Paper:**

Authors introduced approaches aimed at learning invariant predictors across data sources. Rather than using distribution/risk matching schemes as often done by previous work, they propose to train models against mixtures of data points as a means to avoid that models rely on spurious correlations between domain and class labels, since such correlations observed during training might not hold at testing time. The proposed setting uses the idea of mixup to combine data instances in two different schemes: I-combine data points from the same class but different domains, and II-combine data points from the same domain but from different classes.

**Summary Of The Review:**

The paper is well-written, the approach is efficient and observed to work well on a number of benchmarks. However, experiments supporting key claims are lacking, and it's unclear whether the observed improvements in terms of invariance (either at feature- or prediction-level) hold true since no supporting experiments are reported. Contextualization of the proposal relative to past literature also needs improvements since cross-domain data mixup was studied in the past under both settings considered by the authors. The provided discussion on related work does not clarify what and how the authors' proposal improves upon previous work.

---

> ### Author Response · Authors · 2021-11-19
> **Response to Reviewer 4YgJ, part 1/2**
>
> Thanks for your thorough and constructive comments. We have revised our paper with new discussions and experiments based on your comments. We detail our response below and would appreciate it if you could let us know whether our response addresses your concerns.
>
> > The proposal requires assumptions that are not discussed in the manuscript.
>
> Thanks for pointing it out. However, this condition is implied by our model in equation (1) and our assumption on the 9th line on page 8. In the same line, we defined $\Delta$, which is later used in Theorem 1. These assumptions have already been explicitly presented.
>
> ---
>
> > Evaluation of domain invariance representations and prediction-level invariance
>
> **[Representation-level Invariance]** In the initial submission, we have already conducted experiments to evaluate the representation-level invariance in Section 4.5, where measure the invariance by calculating the pairwise KL divergence of domain representations. Compared with other strategies, the results in Table 6 (revised paper) show that representations from different domains are closer together by applying LISA, verifying that LISA leads to stronger representation invariance.
>
> Besides, we did conduct experiments to classify domains using the learned representation on CMNIST. However, all methods achieve an accuracy of 100%. That is what we expect since it is very easy to classify domains with even a little domain-related information.
>
> **[Prediction-level Invariance]** In light of your comment about prediction-level invariance, we conduct a new experiment by calculating the variance of test risks across all domains, where a small variance represents strong prediction-level invariance. The results on CMNIST and MetaShift have been summarized in the following Table and in Section 4.5 of the revised paper. These results indicate that LISA indeed improves prediction-level invariance.
>
> | Dataset | CMNIST | MetaShift |
> | ------ | :-----: | :-----: |
> | ERM | 12.0486 | 1.8824 |
> | Vanilla mixup | 0.2769 | 0.2659 |
> | IRM | 0.0112 | 0.8748 |
> | DomainMix | 0.1674 | 1.1158 |
> | **LISA (ours)** | **0.0012** | **0.2387** |
>
> ---
>
> > Model capacity analysis
>
> Thank you for your suggestion. We conduct experiments to analyze the model capacity by calculating the averaged accuracy on the in-distribution validation set. The performance of ERM, IRM, Coral, LISA on CMNIST, Waterbirds, CelebA, MetaShift are reported in the following:
>
> | Dataset | CMNIST | Waterbirds | CelebA | MetaShift |
> | ------ | ----- | ----- | ----- | ----- |
> | ERM | 73.75% | 91.08% | 91.79% | 93.75% |
> | IRM | 70.28% | 89.32% | 90.53% | 92.85% |
> | Coral | 72.77% | 90.15% | 90.48% | 91.96% |
> | **LISA (ours)** | 73.07% | 89.15% | 90.85% | 94.52% |
>
> Compared with LISA, the results demonstrate that directly adding regularizers on ERM (i.e., IRM and Coral) does limit the model capacity due to the worse in-distribution performance.
>
> ---
>
> > Conclusions from the risk bounds provided in theorems 1 and 2 are a bit unrealistic
>
> The aim of our theoretical analysis is to explain some phenomena distilled from our empirical study and to help us understand why LISA can work. Such a simple model has also been adopted to understand empirical observations in the literature of out-of-distribution robustness [Miller et al. 2021] and in the other subfields of deep learning, such as fairness [Chuang et al. 2021] and adversarial robustness [Schmidt et al. 2018, Carmon et al. 2019].
>
>
> [Miller et al. 2021] Miller, John P., Rohan Taori, Aditi Raghunathan, Shiori Sagawa, Pang Wei Koh, Vaishaal Shankar, Percy Liang, Yair Carmon, and Ludwig Schmidt. "Accuracy on the line: On the strong correlation between out-of-distribution and in-distribution generalization." ICML 2021.
>
> [Chuang et al. 2021] Chuang, Ching-Yao, and Youssef Mroueh. "Fair mixup: Fairness via interpolation." ICLR 2021.
>
> [Schmidt et al. 2018] Schmidt, Ludwig, Shibani Santurkar, Dimitris Tsipras, Kunal Talwar, and Aleksander Madry. "Adversarially Robust Generalization Requires More Data." In NeurIPS. 2018.
>
> [Carmon et al. 2019] Carmon, Yair, Aditi Raghunathan, Ludwig Schmidt, Percy Liang, and John C. Duchi. "Unlabeled data improves adversarial robustness."  In NeurIPS. 2019.

---

> > ### Author Response · Authors · 2021-11-19
> > **Response to Reviewer 4YgJ, part 2/2**
> >
> > > Novelty and discussion with related works
> >
> > In the submission, we have cited four methods that generate more domains and discussed the difference in the first part of the related work. We cite [Shu et al. 2021] and [Wang et al. 2020] in the revised version and these papers also belong to the category of augmenting more domains. As mentioned in the related work, the aim of domain augmentation is to learn a representation that is invariant to the domain, whereas LISA tries to ensure that the final predictions are invariant to domain. Thus, LISA places fewer constraints on the learned representation.
> >
> > In terms of Fair mixup [Chuang et al. 2021], it enables cross-domain mixup regardless of the class information to achieve more fair classification models. In addition, it uses the interpolated data as an explicit regularization term. Compared with Fair mixup, LISA with selection strategy I further restricts the interpolation scope, i.e., mixing samples with the same label, which is specifically suitable in improving out-of-distribution robustness. Moreover, as we mentioned in the second paragraph on page 2 in the original paper, LISA is an implicit method as it applies standard training (without explicit regularization) directly on the interpolated data. In addition, the motivations of Fairmix and LISA are completely different. Fair mixup aims to obtain a fair classifier, while LISA focuses on eliminating the effects of domain-related spurious correlation.
> >
> > [Shu et al. 2021] Shu, Yang, et al. "Open Domain Generalization with Domain-Augmented Meta-Learning." Proceedings of the IEEE/CVF Conference on Computer Vision and Pattern Recognition. 2021.
> >
> > [Wang et al. 2020] Wang, Yufei, Haoliang Li, and Alex C. Kot. "Heterogeneous domain generalization via domain mixup." ICASSP 2020-2020 IEEE International Conference on Acoustics, Speech and Signal Processing (ICASSP). IEEE, 2020.
> >
> > [Chuang et al. 2021] Chuang, Ching-Yao, and Youssef Mroueh. "Fair mixup: Fairness via interpolation." ICLR 2021.
> >
> > ---
> >
> > > On page 2, the setting described was not originally introduced by Koh et al. (2021)
> >
> > Thanks for your comment. We have added the references in the description of distribution shifts in Section 2.
> >
> > ---
> >
> > > One-hot label in equation 2
> >
> > Thanks for pointing it out. We have clarified it in the revised paper.
> >
> > ---
> >
> > > Out-of-distribution definition
> >
> > In the submission, we refer to “out-of-distribution” as the scenario when training and test data are not independent and identically distributed. This definition has been widely used in the machine learning community [Arjovsky et al. 2019; Ye et al. 2021; Koh et al. 2021]. We have revised our paper in the first paragraph of Section 1 to clarify this.
> >
> > [Arjovsky et al. 2019] Arjovsky, Martin, Léon Bottou, Ishaan Gulrajani, and David Lopez-Paz. "Invariant risk minimization." arXiv preprint arXiv:1907.02893 (2019).
> >
> > [Koh et al. 2021] Koh, Pang Wei, Shiori Sagawa, Sang Michael Xie, Marvin Zhang, Akshay Balsubramani, Weihua Hu, Michihiro Yasunaga et al. "Wilds: A benchmark of in-the-wild distribution shifts." ICML 2021.
> >
> > [Ye et al. 2021] Ye, Haotian, Chuanlong Xie, Tianle Cai, Ruichen Li, Zhenguo Li, and Liwei Wang. "Towards a Theoretical Framework of Out-of-Distribution Generalization." NeurIPS 2021.

---

> > > ### Author Response · Authors · 2021-11-27
> > > **We would love to hear back from Reviewer 4YgJ**
> > >
> > > Dear Reviewer 4YgJ,
> > >
> > > We would love to see whether you are still concerning about our empirical analysis. We would really appreciate the opportunity to discuss this further if our response has not already addressed your concern. Many thanks!

---

> > ### Comment · Reviewer_4YgJ · 2021-11-29
> > **Response to authors**
> >
> > Thank you for the detailed response!
> >
> > While the authors addressed some of our concerns, the main ones unfortunately remain.
> >
> > Overall, the paper shows that applications of data mixup can improve out-of-distribution generalization, but it gives no explanation as to why that is. The fact that mixup acts as a regularizer and helps to generalize is already known, thus I would say that the paper lacks in providing new information relative to past literature. Explaining how mixup operates and what it does that improves generalization in the considered setting would make the paper very useful to the community.
> >
> > Specifically, the evaluation reported in table 6 could be further clarified (the KL is measured between which distributions?). However, even if we assume that the results in table 6 support what the authors suggest, in the response, it was mentioned that they were able to train domain classifiers to 100% accuracy, which clearly indicates no domain-invariance is achieved at the feature level. As such, we are left with assuming that invariance is achieved at the prediction level, but that would require stronger empirical evidence to support the conclusion and some explanation as to why mixup would yield such an effect. The authors included experiments measuring the risk variance, but only for CMNIST which makes it unclear whether results would hold in more realistic evaluation conditions and with naturally occurring domain shifts.
> >
> > I also have some other concerns with the presentation; the settings under consideration are not detailed in the introductory sections in that key assumptions are hidden in the theoretical results, while they should be clearly highlighted once the settings are introduced. Domains, for instance, apparently are assumed to correspond to marginal distributions over the input space, but that's never stated in the text. Also, as mentioned in the original review, the assumption that P(y|x) is fixed across domains is a strong one which should be clearly highlighted once the settings under consideration are introduced.
> >
> > The theoretical results also don't seem to add much value given the somewhat unrealistic assumptions they require. The authors mentioned similar assumptions appeared in other settings, but still, in this case it's unclear whether the conclusions would hold in more realistic cases. A discussion highlighting which assumptions would not hold in practice and how that could affect results could be helpful to address that.

---

> > > ### Author Response · Authors · 2021-11-30
> > > **Response to Reviewer 4YgJ's additional concerns**
> > >
> > > Dear Reviewer 4YgJ,
> > >
> > > Thank you for letting us know your additional concerns. We detail our additional response below.
> > >
> > > > the evaluation reported in table 6 could be further clarified (the KL is measured between which distributions?)
> > >
> > > In our initial submission, we have already discussed the distributions used to calculate KL divergence (see Line 8-13 in Section 4.5 in the revised paper). The KL divergence is used to measure the divergence between the distributions of representations from two different domains.
> > >
> > > > in the response, it was mentioned that they were able to train domain classifiers to 100% accuracy, which clearly indicates no domain-invariance is achieved at the feature level.
> > >
> > > We would like to clarify that we train domain classifiers on CMNIST and obtain 100% accuracy for all methods since it is easy to classify domains with even a little domain information in the hidden representations. However, the hidden representations do include domain-invariant information that benefits out-of-distribution robustness. In this case, LISA has demonstrated its promise in learning more invariant representations, which is supported by the results in Table 6. Besides, we further train domain classifiers on Waterbirds and also get very high domain classification accuracy. The results are reported below, and LISA indeed leads to more invariant representations to some extent(i.e., worse domain classification accuracy).
> > >
> > > | Dataset |  WaterBirds |
> > > | ------ |  :-----: |
> > > | ERM | 92.31% |
> > > | Vanilla mixup | 92.57% |
> > > | IRM | 91.23% |
> > > | DomainMix  |  92.35% |
> > > | **LISA (ours)** |  **89.24%** |
> > >
> > > > we are left with assuming that invariance is achieved at the prediction level, but that would require stronger empirical evidence to support the conclusion and some explanation as to why mixup would yield such an effect. The authors included experiments measuring the risk variance, but only for CMNIST which makes it unclear whether results would hold in more realistic evaluation conditions and with naturally occurring domain shifts.
> > >
> > > In our initial response, we have conducted experiments on both CMNIST and MetaShifts, where MetaShifts is a realistic dataset with domain shifts. Besides, we further conduct the experiments on Waterbirds and report the results as follows:
> > >
> > >
> > >
> > > | Dataset |  WaterBirds |
> > > | ------ |  :-----: |
> > > | ERM | 0.2456 |
> > > | Vanilla mixup | 0.1465 |
> > > | IRM | 0.1243 |
> > > | DomainMix  |  0.0995 |
> > > | **LISA (ours)** |  **0.0016** |
> > >
> > > The superiority of LISA further strengthens our conclusion that LISA can improve prediction-level invariance.
> > >
> > > >  the settings under consideration are not detailed in the introductory sections in that key assumptions are hidden in the theoretical results, while they should be clearly highlighted once the settings are introduced. Domains, for instance, apparently are assumed to correspond to marginal distributions over the input space, but that's never stated in the text. Also, as mentioned in the original review, the assumption that P(y|x) is fixed across domains is a strong one which should be clearly highlighted once the settings under consideration are introduced.
> > >
> > > We respectfully disagree with your comment "that key assumptions are hidden in the theoretical results". We did not hide the key assumptions at all.  Our model is clearly presented in equation (1) and our assumptions are explicitly presented on the 9th line on page 8. Your claimed condition "$P(y|x)$ is fixed across domains" is a consequence of our assumptions. The assumptions that we explicitly presented in the paper are sufficient to prove our theoretical results and show the superiority of our proposed method.
> > >
> > > > Contribution of theoretical analysis
> > >
> > > We would like to highlight that our theoretical analysis aims to understand the phenomena in the empirical analysis. The papers listed in our initial response also provide theoretical insights with similar assumptions. In particular, [Miller et al. 2021] also focuses on out-of-distribution robustness. We believe a solid theory can help people to better understand why the proposed algorithm benefits the performance. We will add more discussion in the next version.
> > >
> > > [Miller et al. 2021] Miller, John P., Rohan Taori, Aditi Raghunathan, Shiori Sagawa, Pang Wei Koh, Vaishaal Shankar, Percy Liang, Yair Carmon, and Ludwig Schmidt. "Accuracy on the line: On the strong correlation between out-of-distribution and in-distribution generalization." ICML 2021.

---

### Official Review · Reviewer_iP5p · 2021-11-02

**Correctness:** 4
**Technical Novelty And Significance:** 2
**Empirical Novelty And Significance:** 3
**Recommendation:** 6
**Confidence:** 4

**Main Review:**

Strength:
(1) The authors address a critical point that prevent models from generalization, namely spurious correlation.
(2) The proposed method is simple and easy to implement, and the empirical results are within expectation.

Weakness:
(1) Maybe the biggest concern is the contribution over previous work. The proposed method can be seen as a heuristic extention of mixup. Though simple and easy to follow, the contribution is marginal.
(2) There ara growing trends on investigating OOD generalization under missing domain label, it is better to at least include such work (e.g. [1]) for discussion.
(3) There is a hyper-parameter p_sel that controls the probability of performing different strategy, is there a rule of thumb or we need to tune it for every task?
[1] Qiao, F., Zhao, L., & Peng, X. (2020). Learning to Learn Single Domain Generalization. In CVPR.

**Summary Of The Paper:**

The paper considers the model robustness under distribution shift brought by domains and subpopulations. Specifically, based on the interpolation scheme in mixup, the authors propose two selection strategies to perform data augmentation, aim at eliminating the spurious correlations and learning an invariant representation.

**Summary Of The Review:**

This paper address the spurious correlation by augmenting the data via interpolation. Although intuitions are provided and empirical effectiveness is illustrated accordingly, the contributions over previous work (e.g. mixup) are marginal.

=========After Response==========

The response from authors has addressed my major concerns, so I raise my score accordingly.

---

> ### Author Response · Authors · 2021-11-19
> **Response to Reviewer iP5p**
>
> Thanks for your valuable comments. We have revised our paper according to your constructive suggestions. We would appreciate it if you could let us know whether our response addresses your concerns.
>
> > Maybe the biggest concern is the contribution over previous work. The proposed method can be seen as a heuristic extension of mixup
>
> We agree the method is a conceptual extension of mixup. However, LISA mainly focuses on out-of-distribution robustness, and the performance gains compared to vanilla mixup are significant. Further, we provide extensive empirical and theoretical analyses that show and help understand the effectiveness of LISA compared to vanilla mixup. Therefore, we believe that the contribution over the previous work is significant.
>
> We also would like to note that although mixup is very popular in practice, its theoretical understanding is still limited. To the best of our knowledge, our paper provides the first theoretical analysis of how mixup (with and without the selection strategies) affects mis-classification error across different domains.
>
> ---
>
> > There are growing trends on investigating OOD generalization under missing domain label, it is better to at least include such work (e.g. [1]) for discussion.
>
> Thanks for your suggestion. We have added the discussion about out-of-domain generalization without domain information in the first paragraph of Section 6.
>
> ---
>
> > How to tune p_sel for each task?
>
> As we discussed in the second paragraph of page 5 and the first paragraph of page 6 in the initial submission, the choice of $p_{sel}$ depends on the number of domains and the relation between domain information and spurious correlations. Empirically, using selection strategy I only brings much more benefits when there are more domains, and/or the domain information can not fully reflect the spurious correlations. LISA with selection strategy II can benefit the performance when domain information is highly spuriously correlated with the label, where we find a balanced ratio (i.e., $p_{sel}=0.5$) performs the best. We have moved this information to the last paragraph of Section 3 to clarify it.

---

### Official Review · Reviewer_VX2N · 2021-11-03

**Correctness:** 3
**Technical Novelty And Significance:** 2
**Empirical Novelty And Significance:** 3
**Recommendation:** 5
**Confidence:** 4

**Main Review:**

the idea is quite simple and intuitively reasonable, and empirical results seem extensive and significant, and theoretically justified to some extent

some of the results analysis is a bit confusing to me
(1) in 4.1 "evaluating robustness to domain shifts", the best strategy was to always mixup same label with different domains, and the potential reason given is that the datasets actually have weak or even no spurious correlation between domain and label. Two questions follow:
(a) From early text, seems both selection strategies are motivated by the spurious correlation, but here why do we still observe advantage over ERM or vanilla mixup? It would be great if you could clarify the different motivations (if any) of the two selection strategies
(b) the reasoning about "weak or no spurious correlation" seems to be contradicting with claim in 4.3, where it's stated that "compared with vanilla mixup, ...LISA...improve the OOD robustness by canceling out the spurious correlations..". or did I misunderstand something? Is it easy to quantify such spurious correlation? if so, why not present the actual correlation metrics for these datasets?

(2) for the ablation study, I think a more convincing way would be: first test LISA and mixup on a dataset that is known to have NO spurious correlations, then we expect neutral results; then test them on a data that is known to have spurious correlations, and we could give quantitative metrics of such correlations if possible, and show LISA is better than mixup; further more, the stronger the correlation, the more advantage LISA has. Is that what you're trying to demonstrate here?

(3) In table 8, vanilla mixup shows much worse performance than ERM in terms of learning invariant representations under the defined metric, which doesn't seem quite reasonable to me. Shouldn't we expect the contrary?



Another minor question: In Theorem 1, is p the dimension of x? If so it's better to state that explicitly in the theorem instead of relying on readers to go back to text and only to find in the superscript notation.

**Summary Of The Paper:**

This paper propose a mixup-style data augmentation method under the data distribution shift context. In particular, data distributions are formulated as mixture of distributions (i.e., domains), and two distribution shift scenarios are considered: (1) domain shift, where the test domain and train domain are disjoint. (2) subpopulation shift, where test distribution has different mixture proportion than train distribution. It's  assumed that domain identification spuriously correlates with labels. To tackle this problem, this paper proposes two mixup strategies: (I) mixup two examples with same label but different domains; (II) mixup two examples with same domain but different labels. It's claimed that such mixup could cancel out the spurious correlations. Extensive experiments on a variety of datasets show its superiority compared to empirical risk minimization (ERM) and alternative data augmentation methods. The paper further provide theoretical analysis that under certain conditions, the proposed method has asymptotically smaller worst case classification errors than ERM and vanilla mixup.

**Summary Of The Review:**

the idea is quite simple and intuitively reasonable, and empirical results seem extensive and significant, and theoretically justified to some extent, but the results analysis and some experimental design could be more insightful or improved.

---

> ### Author Response · Authors · 2021-11-19
> **Response to Reviewer VX2N, part 1/2**
>
> Thank you for your constructive feedback. We have improved our paper according to your constructive comments and we addressed your questions below. Please kindly let us know if your questions are addressed.
>
> >（a) From early text, seems both selection strategies are motivated by the spurious correlation, but here why do we still observe advantage over ERM or vanilla mixup? It would be great if you could clarify the different motivations (if any) of the two selection strategies (b) the reasoning about "weak or no spurious correlation" seems to be contradicting with claim in 4.3, where it's stated that "compared with vanilla mixup, ...LISA...improve the OOD robustness by canceling out the spurious correlations..". or did I misunderstand something?
>
> -  We apologize for the confusion. Though both selection strategies in LISA aim to reduce the effect of spurious correlations, we first clarify the different motivations between them below. We also revised Section 3, Section 4.1 to clarify it.
>
>    * Selection strategy I aims to directly cancel out the spurious correlation. If the domain information fully reflects spurious correlation, interpolating samples with the same label but different domains can effectively cancel out the spurious correlation. However, if the current domain information does not fully reflect the spurious correlations, directly applying selection strategy I even without using the given domain information yields the best performance, which has been initially discussed in Appendix A.1.2 and moved to the "Results" paragraph of Section 4.1 and Section 3.
>
>    * Selection strategy II, instead, focuses on actively building domain-independent prediction models, where domain information is required to be highly spuriously correlated with the labels. Otherwise, selection strategy II may not be suitable.
>
> - Then, we clarify that there are indeed spurious correlations in datasets with domain shifts, while the current domain information may not fully reflect the spurious correlations. As we mentioned in Section 4.1, in Camelyon17-wilds dataset (Koh et al., 2021), the presence of tumor tissue (i.e., label) mainly depends on the demographic of patients (e.g., race, gender), which shows no significant difference across hospitals (i.e., domain information). This could also explain why ERM, which does not consider domain information, outperforms other baseline models under the domain shifts scenario. Under this setting, we directly use LISA-I without domain information to cancel out the spurious correlations, i.e., $p_{sel}=1.0$. We have revised our paper in the "Results" paragraph of Section 4.1 to clarify it.
>
> ---
>
> > Ablation study on datasets without spurious correlations
>
> According to your suggestions, we conduct new experiments on datasets without spurious correlations. To be more specific, we balance the number of samples for each group under the subpopulation shifts scenario. The results of ERM, vanilla mixup, and LISA on CMNIST, Waterbirds, and CelebA are reported in the following table:
>
> | Dataset | ERM | Vanilla Mixup | LISA |
> | ------ | :-----: | :-----: | :-----: |
> | CMNIST | 73.67% | 74.28% | 73.18% |
> | Waterbirds | 88.07% | 88.23% | 87.05% |
> | CelebA | 86.11% | 88.89% | 87.22% |
>
> The above results show that LISA performance is similar to ERM when datasets do not have spurious correlations. If there exists any spurious correlation, LISA significantly outperforms ERM. Another interesting finding is that vanilla mixup outperforms LISA and ERM without spurious correlations, while LISA achieves the best performance with spurious correlations. This finding strengthens our conclusion that the performance gains of LISA are caused by eliminating spurious correlations rather than data augmentation. We have revised Section 4.3 and Appendix A.3 to include the new results and analysis.
>
> ---
>
> > the stronger the correlation, the more advantage LISA has. Is that what you're trying to demonstrate here?
>
> Yes, in Section 4.4, we have analyzed the strength of spurious correlation and the performance gains of LISA. The results corroborate our hypothesis that LISA leads to more improvements with the spurious correlations are stronger

---

> > ### Author Response · Authors · 2021-11-19
> > **Response to Reviewer VX2N, part 2/2**
> >
> > > Comparison between vanilla mixup and ERM in Table 6 (revised paper)
> >
> > This scenario is what we expected. In subpopulation shifts (e.g., CMNIST), vanilla mixup actually makes the learned representations less invariant. For example, imagine the toy example with two data groups and an imbalance factor 0.9, which means the proportion between the majority and minority group is 9:1. With vanilla mixup, the percentage of the pure majority/minority groups is 81%/1%, and there are 18% mixed samples from both the majority group and minority group. According to the proportion between the pure majority/minority groups, the model is still biased towards the majority group. The biased model will then make the mixed samples biased to the majority group since they have the information from the majority group, which essentially strengthens the ratio of the majority group and worsen the performance.
> >
> > ---
> >
> > > Is p the dimension of x?
> >
> > Thanks for your suggestion. Yes, p is the dimension of x. We have revised the paper  to include the definition in the theorem.

---

> > > ### Author Response · Authors · 2021-11-27
> > > **We would love to hear back from Reviewer VX2N**
> > >
> > > Dear Reviewer VX2N,
> > >
> > > We would love to hear back about whether you still have concerns about our experiments and ablation study. We are more than happy to further provide explanations or clarifications if more is needed. Many thanks!

---

> > ### Comment · Reviewer_VX2N · 2021-12-01
> > **still a bit confused**
> >
> > Thanks for the detailed response!
> >
> > I have read your response and revision in section 4.1, I don't feel it becomes more clear, where the first "key finding" says there is no spurious correlation in Camelyon17-wilds, but the second "key finding" says "improving OOD robustness by canceling out spurious correlations with augmentation."
> > What does "existing domain information may not fully reflects the spurious correlation" mean? I thought it means no spurious correlation.
> >
> > Another thing I just noticed is in Table 1 some are using worst acc but some are using average acc, but seems no explanation on why different metrics are used for different datasets?
> >
> > Another concern I just realized is, as reviewer iP5p also raised: seems $p_{sel}$ is not a typical ML hyper parameter that we can't choose with cross validation? and I don't see the response to this question is an actionable algorithm on how to choose this parameter.

---

> > > ### Author Response · Authors · 2021-12-01
> > > **Response to Reviewer VX2N's Additional Concerns**
> > >
> > > Dear Reviewer VX2N,
> > >
> > > Thank you for letting us know your additional concerns.
> > >
> > > > I have read your response and revision in section 4.1, I don't feel it becomes more clear, where the first "key finding" says there is no spurious correlation in Camelyon17-wilds, but the second "key finding" says "improving OOD robustness by canceling out spurious correlations with augmentation." What does "existing domain information may not fully reflects the spurious correlation" mean? I thought it means no spurious correlation.
> > >
> > > The spurious correlations do exist in these datasets. However, the domain information may not fully reflect the spurious correlation, i.e., some spurious correlations may be unobserved or challenging to capture. That's why we get the first finding. LISA is still capable of eliminating these unobserved spurious correlations, leading to the second finding. We will clarify it in the next version.
> > >
> > > > Another thing I just noticed is in Table 1 some are using worst acc but some are using average acc, but seems no explanation on why different metrics are used for different datasets?
> > >
> > > In our experiments, we follow the setting in the WILDS benchmark [Koh et al. 2021], which uses different metrics for different datasets with detailed explanations.
> > >
> > > > Another concern I just realized is, as reviewer iP5p also raised: seems $p_{sel}$ is not a typical ML hyper parameter that we can't choose with cross validation? and I don't see the response to this question is an actionable algorithm on how to choose this parameter.
> > >
> > > $p_sel$ is a typical hyperparameter, and we can definitely use cross-validation to choose this hyperparameter. In our response to reviewer iP5p, we provide instruction about how to choose $p_{sel}$ roughly. Cross-validation is always a good way to get concise value.

---

> > > > ### Comment · Reviewer_VX2N · 2021-12-05
> > > > **cross-validation error doesn't reflect test error**
> > > >
> > > > thanks for the explanation.
> > > >
> > > > "spurious correlations do exist in these datasets. However, the domain information may not fully reflect the spurious correlation, i.e., some spurious correlations may be unobserved or challenging to capture" -- Are you saying "spurious correlations do exist in these datasets" is purely based on speculation without real data points? If so I'm afraid the argument is not that strong.
> > > >
> > > > I don't quite follow how you could use cross-validation to select the $p_{sel}$ parameter. Given that the algorithm is targeting cases where test distribution is different from training distribution, will cross-validation performance still a good estimate of the true test error? At least not naive cross validation. You might have to do some special handling to the validation set to simulate the test set?

---

> > > > > ### Author Response · Authors · 2021-12-08
> > > > > **Response to Reviewer VX2N's additional questions about corss-validation**
> > > > >
> > > > > Dear Reviewer VX2N,
> > > > >
> > > > > Thank you for letting us know your additional concerns.
> > > > >
> > > > > > Are you saying "spurious correlations do exist in these datasets" is purely based on speculation without real data points? If so I'm afraid the argument is not that strong.
> > > > >
> > > > > Our claim is built on the empirical finding, where in-distribution test accuracy is significantly higher than out-of-distribution test accuracy. Take RxRx1 as an example (see Table 12), the in-distribution test accuracy of ERM is 35.9 $\pm$ 0.4%, whereas the out-of-distribution test accuracy is 29.9 $\pm$ 0.4%. If there is no spurious correlation, the in-distribution test accuracy should be similar to the out-of-distribution test accuracy. Thus, there do exist some spurious features that correlate with the label.
> > > > >
> > > > > > Given that the algorithm is targeting cases where test distribution is different from training distribution, will cross-validation performance still be a good estimate of the true test error? At least not naive cross validation. You might have to do some special handling to the validation set to simulate the test set?
> > > > >
> > > > > Cross-validation indeed works in our setting, where the validation set could be regarded as out-of-distribution data. For example, in Camelyon-17, we use the samples from three hospitals for training. In cross-validation, we can randomly pick two hospitals for training, and the rest is used for validation.

---

### Author Response · Authors · 2021-11-19
**Overall Summary of Changes**

We sincerely thank all reviewers for their constructive feedback. We summarize all major changes in the revised paper below. All changes are in red text.

1. We revised the “Introduction” and “Preliminaries” to include the definition of “out-of-distribution” and add more references.

2. In Section 3, we clarified the motivations and application scopes of selection strategies I and II in LISA.

3. In Section 4.1, We clarified the explanations and findings of domain shifts.

4. In Section 4.3 and Appendix A.3, we add a new experiment on datasets without spurious correlation.

5. In Section 4.5, we add a new experiment to analyze the prediction-level invariance of LISA and other baselines.

6. We revised the related work (Section 6) to include more references and discussions.

---

### Decision · Program_Chairs · 2022-01-20

**Decision:**

Reject

**Comment:**

The manuscript focuses on model robustness under distribution shift, specifically domain shifts and subpopulation shifts. Domain shift is where the test domain and train domain are disjoint. Subpopulation shift is where test distribution has different mixture proportion than train distribution. The assumption is that domain identification spuriously correlates with labels. The proposed framework learns an invariant representation by using mixup strategies and interpolates samples either with the same labels but different domains or with the same domain but different labels to. Experiments are performed on a variety of domain shift and subpopulation shift benchmarks, and results showed that the proposed framework is better than empirical risk minimization (ERM) and alternative data augmentation methods. Theoretical analysis is also provided and it is shown that, under certain conditions, the proposed framework has asymptotically smaller worst case classification errors than ERM and vanilla mixup.

Reviewers agreed on several positive aspects of the manuscript, including:
1. The manuscripts addresses a critical point that prevent models from generalization, namely spurious correlation;
2. The proposed method is simple and easy to implement, and the empirical results are within expectation.

Reviewers also highlighted several major concerns, including:
1. Different recent approaches introduce methods that use some sort of mixup across domains in similar settings;
2. Ablation study on datasets without spurious correlations are missing;
3. Evaluation of domain invariance representations and prediction-level invariance needs clarifications;

Authors clarified different motivations of the two selection strategies in relation to spurious correlation between domains and labels, and provided an ablation study on datasets with no spurious correlation. Post-rebuttal, reviewers stayed with borderline ratings, and they have suggested further improvements: improving results analysis and the conclusion that “existing domain information may not fully reflect the spurious correlation”, understanding the implication and the reasons that invariance is achieved at the prediction level instead of at the representation level despite the original goal is to learn an invariant representation, and improving presentation of the manuscript including settings and assumptions.